# Auditory cortex neurons that encode negative prediction errors respond to omissions of sounds in a predictable sequence

**Amit Yaron**[1], **Tomoyo Shiramatsu-Isoguchi**[2], **Felix B. Kern**[1], **Kenichi Ohki**[1,3,4‡],
**Hirokazu Takahashi**[1,2‡], **Zenas C. Chao**[1‡*]

**1** International Research Center for Neurointelligence (WPI-IRCN), UTIAS, The University of Tokyo, Tokyo, Japan, **2** Department of Mechano-Informatics, Graduate School of Information Science and Technology, The University of Tokyo, Tokyo, Japan, **3** Department of Physiology, Graduate School of Medicine, The University of Tokyo, Tokyo, Japan, **4** Institute for AI and Beyond, The University of Tokyo, Tokyo, Japan

‡ KO, HT, and ZCC are joint senior co-authors on this work.
* zenas.c.chao@gmail.com

## Abstract

Predictive coding posits the brain predicts incoming sensory information and signals a positive prediction error when the actual input exceeds what was predicted, and a negative prediction error when it falls short of the prediction. It is theorized that specific neurons encode the negative prediction error, distinct from those for the positive prediction error, and are linked to responses to omitted expected inputs. However, what information is actually encoded by omission responses remains unclear. This information is essential to confirm their role as negative prediction errors. Here, we record single-unit activity in the rat auditory cortex during an omission paradigm where tone probabilities are manipulated to vary the prediction content. We identify neurons that robustly respond to omissions, with responses that increase with evidence accumulation and directly correlate with tone predictability—key characteristics suggesting their role as negative prediction-error neurons. Interestingly, these neurons showed selective omission responses but broad tone responses, revealing an asymmetry in error signaling. To capture this asymmetry, we propose a circuit model composed of laterally interconnected prediction-error neurons that qualitatively reproduce the observed asymmetry. Furthermore, we demonstrate that these lateral connections enhance the precision and efficiency of prediction encoding across receptive fields, and that their validity is supported by the free energy principle.

## Introduction

Predictive coding, a theoretical framework that explains how the brain processes sensory information, posits that the brain generates predictions based on prior experiences and compares them with actual sensory inputs [1–5]. The discrepancy

**Data availability statement:** Raw experimental data are available from Zenodo: https://doi.org/10.5281/zenodo.15531781 MATLAB analysis code is available from Zenodo: https://doi.org/10.5281/zenodo.15531880 Python model code is available from Zenodo: https://doi.org/10.5281/zenodo.15532335 Individual numerical data underlying Figs 2D, 4A, S2A, S2B and S8 are provided as Supporting information.

**Funding:** This work was supported by the World Premier International Research Center Initiative (WPI) of MEXT, Japan (https://www.jsps.go.jp/english/e-toplevel/) (awarded to A.Y. and Z.C.C.); JSPS KAKENHI (https://www.jsps.go.jp/english/e-grants/) (23K14298 awarded to A.Y.; 23H03465, 23H03023, and 24H01544 awarded to H.T.; 23H04336 awarded to T.S.); AMED (https://www.amed.go.jp/en/index.html) (24wm0625401h0001 awarded to H.T.); JST (https://www.jst.go.jp/EN/) (JPMJPR22S8 awarded to T.S.); the Asahi Glass Foundation (https://www.af-info.or.jp/en/) (awarded to H.T.); and the Secom Science and Technology Foundation (https://www.secomzaidan.jp/) (awarded to H.T.). The funders had no role in study design, data collection and analysis, decision to publish, or preparation of the manuscript.

**Competing interests:** The authors have declared that no competing interests exist.

**Abbreviations:** AAF, anterior auditory field; AEPs, auditory evoked potentials; CSD, current source density; FRA, frequency response area; LFP, local field potential; LIF, leaky integrate-and-fire; MMN, mismatch negativity; OSI, omission selectivity index; PEONs, probability encoding omission neurons; PSTHs, peri-stimulus time histograms; SOA, stimulus onset asynchrony; TSI, tone selectivity index; VAF, ventral auditory field; WCSS, within-cluster sum of squares.

between the predicted and actual input, known as the prediction error, is used to update the brain's internal models. This process is thought to occur through canonical circuits, hierarchically organized neural networks where top–down predictions and bottom–up prediction errors create a feedback loop that allows the brain to refine its models.

Recent work suggested that positive prediction errors, occurring when the actual input exceeds the predicted input, and negative prediction errors, occurring when the actual input is less than the predicted input, are calculated in separate populations of neurons within these canonical circuits [6–9]. While extensive research has investigated positive prediction errors using the oddball paradigm and its variations [10–18] these studies face significant limitations. Specifically, oddball responses consist of more than just positive prediction errors. First, they also contain sensory signals, which are often hard to disentangle. Additionally, a deviant stimulus also represents the omission of the standard stimulus, thereby generating a negative prediction error, which further complicates the interpretation. In contrast, responses to omitted expected stimuli do not contain sensory input and could directly represent negative error signals, potentially offering a crucial step in elucidating the computation of prediction errors. Omission responses have been observed in human studies [19–24] and more recently in single neurons within the visual and auditory cortices [7,9,25–28]. However, omission responses could arise from processes other than predictive coding and may not necessarily represent negative prediction errors. Therefore, it is crucial to characterize the information encoded in these responses to confirm their functional role and investigate how they might arise from the interactions between prediction and sensory signals, an area that remains largely unexplored.

To address this, we investigated how neurons in the rat auditory cortex respond to the omission of expected auditory stimuli while manipulating their predictability. We use high-resolution extracellular recordings to simultaneously record activity from a large population of neurons across multiple cortical layers, while presenting tone sequences where the probability of tone presentations was systematically varied. We identified a subset of auditory neurons that exhibit two key characteristics of negative prediction-error encoding. First, the neurons displayed omission responses whose intensity was directly correlated with the statistical likelihood of specific omitted tones, demonstrating a quantitative representation of the prediction error. Second, these neurons exhibited omission responses that built up from trial to trial, reflecting predictions established by the accumulation of evidence through repeated exposure to the stimuli. These neurons, which we named "Probability Encoding Omission Neurons" (PEONs), were primarily found in the granular and supragranular layers of the auditory cortex and were distributed across different auditory fields including primary auditory cortex (A1), ventral auditory field (VAF), and anterior auditory field (AAF), with a significantly higher proportion observed in A1.

Interestingly, while PEONs showed selective responses to omissions, they responded broadly to tones, highlighting an asymmetry between the processing of top–down predictions and bottom–up sensory signals. To explain these dynamics, we propose a novel circuit model that includes separate negative and positive error

calculations within narrowly tuned auditory streams, with prediction error neurons laterally interconnected between adjacent streams. This architecture not only accounts for the observed asymmetry and probability encoding in sensory processing but also demonstrates how lateral connections can enhance the precision and efficiency of prediction-encoding, a concept supported by the free energy principle [1,3]. Our findings provide empirical evidence for the existence of negative prediction error neurons in the auditory cortex and propose a circuit-level hypothesis for how these errors might be computed and integrated within cortical microcircuits.

## Results

### Neuronal responses during auditory omission paradigm

In this study, we focused on the right auditory cortex of Urethane-anesthetized rats, utilizing extracellular recordings to monitor responses to auditory stimuli. Across 18 electrode penetrations involving 10 rats, we successfully captured the activity of 990 single units. For each rat, we first used a surface microelectrode array (NeuroNexus, Ann Arbor, MI, USA) (Fig 1A) to map the various parts of the auditory cortex (Fig 1B) [29–31]. We then used a Neuropixels electrode array to record different layers of the chosen area for each penetration (Fig 1C).

To explore the neural response to auditory stimuli and their omission, we devised eight experimental conditions (Fig 1D). Each condition featured an oddball sequence of two tones with distinct probabilities, along with a consistent

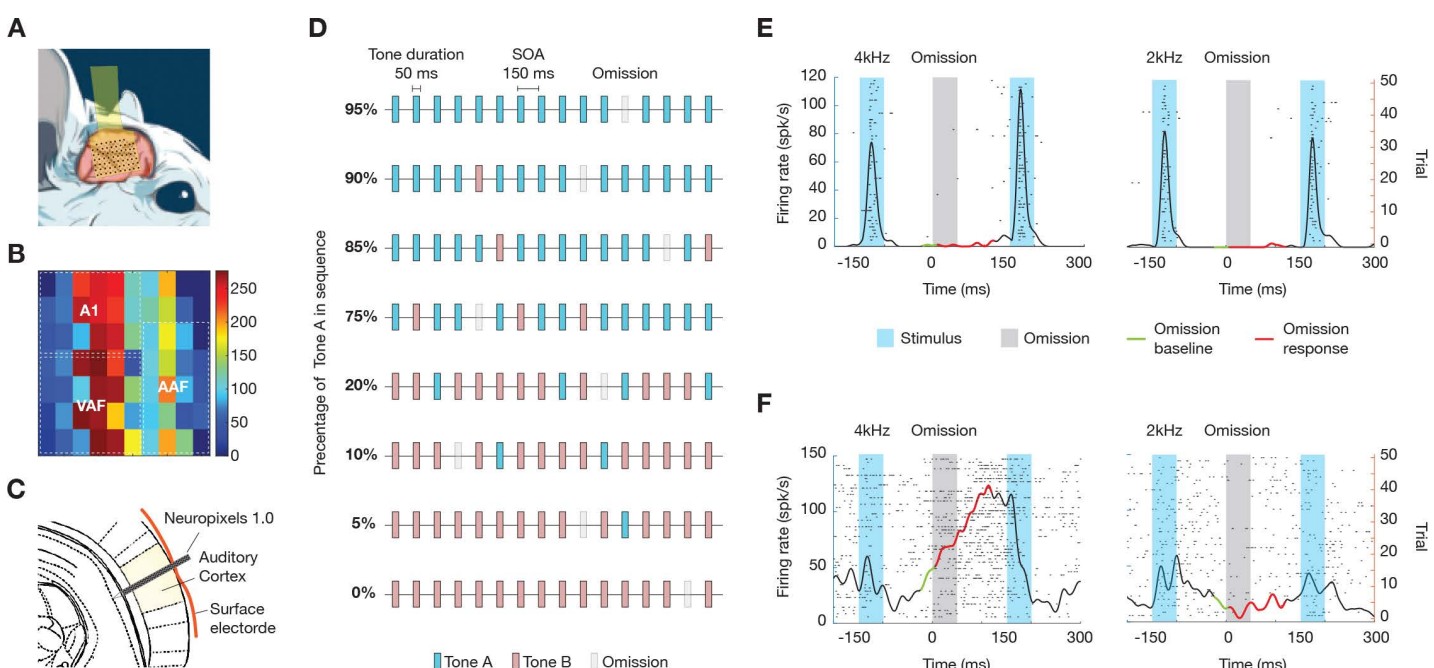

**Fig 1. Neuronal recording and experimental paradigm. (A)** Surface microelectrode array used to map the auditory cortex in urethane-anesthetized rats. **(B)** Representative auditory-evoked potentials in the cortex elicited by click stimuli. The color bar represents P1 amplitude (see "Methods"). A1, AAF, and VAF represent the primary auditory cortex, anterior auditory field, and ventral auditory field, respectively. **(C)** Neuropixels electrode array inserted through the holes on the surface array into the mapped regions to record neural activity across different cortical layers. **(D)** Experimental design featuring eight conditions with two distinct tones **(A and B)** and a 5% omission rate, presented in random order. Each condition comprised 1,000 stimulus items, varying the probability of Tones A and B to achieve different predictability levels. **(E)** Example of a neuron that responds to tones but not to omissions. The raster plot shows spikes (black dots) with time (ms) on the *x*-axis and trials on the *y*-axis. The PSTH displays firing rate over time, with tone stimuli (blue shading) and omissions (gray shading). The black line represents the firing rate, the red line shows omission responses, and the green line indicates the omission baseline. **(F)** Example of a neuron that selectively responds to the omission of specific tones. The same representation is used as in panel E.

occurrence of tone omissions at a rate of 5%. To create varied prediction contents while maintaining a fixed omission rate, we utilized two tones and controlled their relative predictability. We selected two distinct pure tones (denoted as Tones A and B), one octave apart, that consistently elicited responses from most neurons for each penetration (see details in "Methods"). The sequences were systematically designed but presented in a random order. Each sequence comprised 1,000 stimulus items, including omissions, and featured Tones A and B at varying probabilities to achieve distinct levels of predictability. The specific ratios of Tones A and B were carefully controlled as follows: 95% Tone A and 0% Tone B (exclusively Tone A), 90% and 5%, 85% and 10%, and 75% and 20%. This pattern continued, adjusting further to 20% and 75%, 10% and 85%, 5% and 90%, until reaching 0% and 95% (exclusively Tone B).

We observed diverse patterns of neural responses to omissions. Fig 1E and 1F illustrate responses from three exemplary neurons to tone omissions during two distinct tone sequences: one with 95% Tone A and 5% omissions, and the other with 95% Tone B and 5% omissions. Some neurons exhibited strong responses when either tone was presented but did not respond to their omission (Fig 1E), and some neurons displayed selective responses to the absence of specific tones, with the intensity and nature of these responses varying depending on which tone was omitted (Fig 1F).

## Omission responses encode tone probability

To determine whether omission responses encode negative prediction errors, we investigated how neurons' responses to omissions were modulated by the predictability of the omitted stimuli. For neurons encoding negative prediction errors, their response should correlate with the content of the prediction. In our experimental design, the content is determined by the probabilities of the two tones, which is equivalent to the probability of one tone, since their total is fixed at 95%. Therefore, we first evaluated which neurons' omission response correlated with the probability of tone presentation. The omission response was defined as the neuron's mean firing rate during a 5–120 ms window after the omission, with a local baseline subtracted on a per-trial basis. This baseline was calculated as the mean firing rate from –24 to 5 ms relative to the expected tone onset, capturing the neuron's state just before the tone was omitted [32]. We then computed the Spearman's correlation between each neuron's average omission response under each condition and the corresponding probability of tone presentation (either Tone A or B). For each neuron, the tone whose omission response increased with higher probability was designated the "omission preferred" ($O_P$) tone, while the other was termed the "omission non-preferred" ($O_{NP}$) tone. Under this definition, the correlation coefficients measured against the $O_P$ tone were all positive, unlike those measured against the $O_{NP}$ tone, which would be negative.

Next, we examined whether distinct groups exist based on how strongly each neuron's omission response correlated with predictability of the $O_P$ tone. To achieve this, we performed *k*-means clustering on the correlation coefficients and used an unbiased elbow analysis to determine the optimal number of clusters (see details in "Methods"). The elbow analysis revealed that it is optimal to cluster the data into two distinct neuronal subpopulations (see the elbow analysis in S1 Fig): one comprising 771 neurons with little or no modulation of omission responses by tone probability (mean Spearman's $\rho = 0.048$), and a second comprising 219 neurons exhibiting a robust positive correlation (mean $\rho = 0.18$) (Fig 2A).

Fig 2B illustrates the omission responses of two example neurons across eight different conditions. These range from scenarios with 0% $O_P$ tone (95% $O_{NP}$ and 5% omission) to scenarios with 95% $O_P$ tone (0% $O_{NP}$ and 5% omission). The peri-stimulus time histograms (PSTHs), aligned to the expected onset of the omitted tone, have been baseline-subtracted to isolate the omission response. Neuron 1 (from Cluster 1 in Fig 2A) exhibited only a weak correlation (Spearman's $\rho = 0.12$), while Neuron 2 (from Cluster 2) showed a pronounced increase in omission response as the probability of the $O_P$ tone increased (rho = 0.36).

To further examine these PEONs without introducing double-dipping biases, we conducted a split-data analysis by dividing trials into two independent subsets (odd and even trials, denoted as ODD and EVEN trials) to ensure the robustness of our analyses. All PEONs were identified using only one half (e.g., ODD) and subsequently tested on the complementary half (e.g., EVEN), ensuring that selection and evaluation were fully independent. Using the ODD trials, we

PLOS Biology

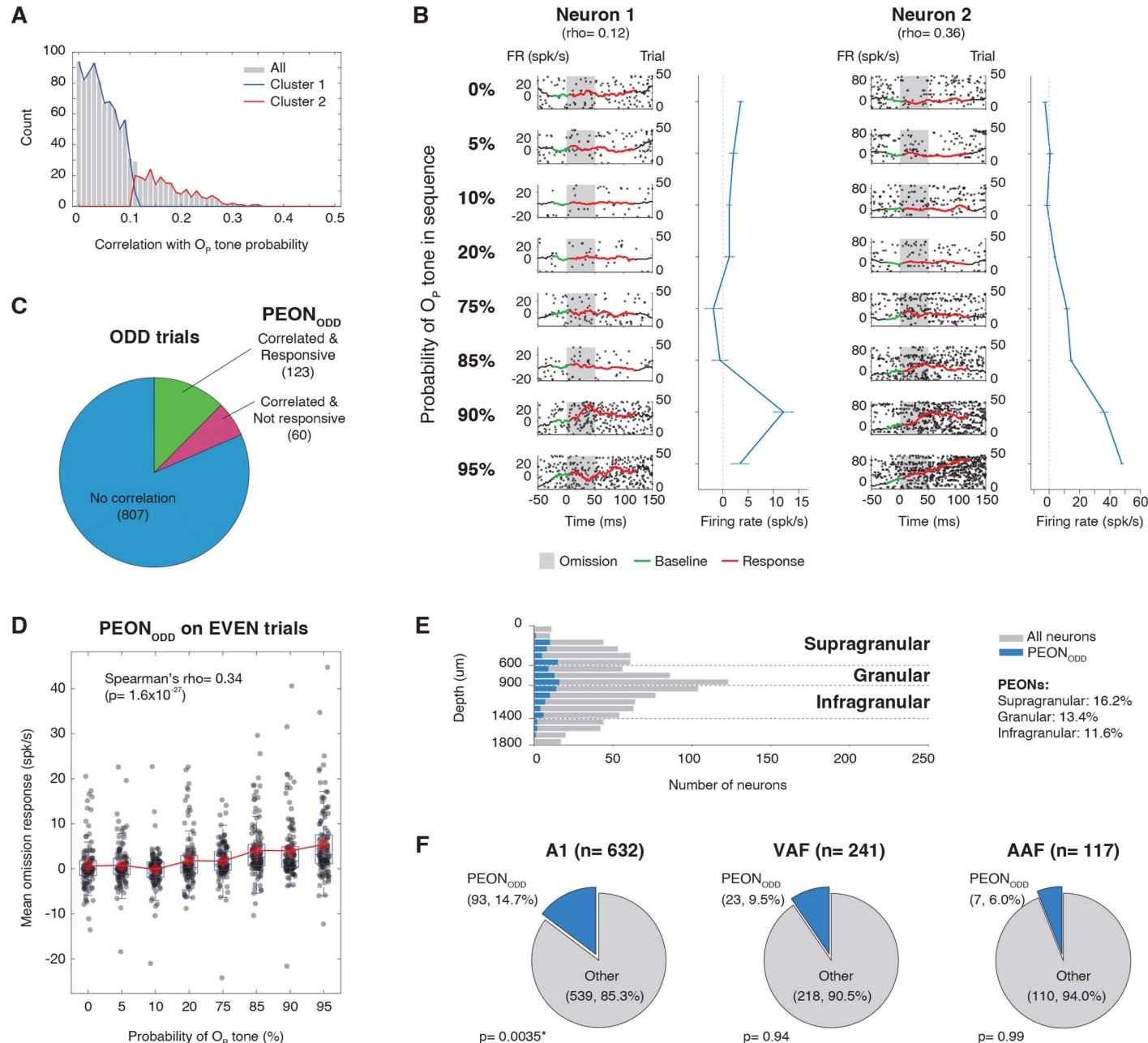

**Fig 2. Split-Data analysis for probability encoding.** **(A)** Distribution of Spearman's correlation coefficients (rho) between each neuron's omission response and OP tone probability. The histogram (gray bars) shows in how many neurons each correlation magnitude occurs, and the two overlaid curves represent separate clusters identified via *k*-means clustering analysis. **(B)** Omission responses from two example neurons across eight conditions ranging from 0% to 95% probability of the $O_P$ tone. Each panel displays single-trial spike raster plot (dots, right *y*-axis) and a peristimulus time histogram (black line, left *y*-axis) aligned to the expected onset of the omitted tone (time = 0 ms). The red line shows omission responses, and the green line indicates the omission baseline. The omission period is indicated by gray shading, and a local baseline is shown in green. The right plots in each row depict the mean omission response (blue curve) for that probability condition, with the *x*-axis showing $O_P$ tone percentage and the *y*-axis showing firing rate. **(C)** Proportion of neurons classified using ODD trials as: PEONs (neurons showing both significant Spearman's correlation between omission response and $O_P$ tone probability plus significant omission response in Wilcoxon sign-rank test when comparing omission responses to baseline in conditions where $O_P$ tone was standard), neurons with correlation only (showing significant correlation but no significant omission response), and neurons with no significant correlation. **(D)** Box plot summarizing omission responses for PEON$_{ODD}$, tested on EVEN trials. The *x*-axis represents the $O_P$ tone probability (0%–95%), and the *y*-axis indicates mean baseline-subtracted firing rate in response to the omission, calculated over a 5–120 ms window after the expected tone onset. Boxes correspond to the interquartile range (IQR) with a horizontal line marking the median. Whiskers extend to 1.5× IQR. Individual data points

are shown as overlaid dots, with means at each probability level marked by red diamonds connected by a dashed line. The text annotation shows the Spearman's rank correlation coefficient (rho) and associated *p*-value, quantifying the strength and statistical significance of the monotonic relationship between probability and neural response. Underlying data are provided in S1 Data. **(E)** Laminar distribution of the recorded population (gray bars) and the neurons classified as PEON$_{ODD}$ (blue bars), shown as a function of cortical depth (*y*-axis). Horizontal dashed lines indicate the approximate boundaries between supragranular (0–600 μm), granular (600–900 μm), and infragranular (900–1,400 μm) layers, based on current source density analysis. **(F)** Distribution of PEONs across auditory cortical fields. Three pie charts illustrate the proportion of PEONs (blue slice) vs. other recorded neurons (gray slice) within A1 (left), VAF (middle), and AAF (right). Labels on each slice indicate the neuron category, the absolute count, and the percentage relative to the total number of neurons recorded in that specific field (indicated in the title of each chart). The *p*-value shown below reflects the result of a bootstrap analysis comparing the observed PEON count in each area to a simulated null distribution; the asterisk denotes statistical significance ($p < 0.05$).

identified 123 PEONs according to two criteria: (1) a significant correlation ($p < 0.05$) between their omission responses and the probability of the corresponding $O_P$ tone, as verified by Spearman's correlation test; and (2) a significant omission response ($p < 0.05$, Wilcoxon sign-rank test) to this $O_P$ tone in the four conditions where it was presented as standard (75%, 85%, 90%, and 95%), when the $O_P$ tone is most predictable and omission responses should be greatest (see details in "Methods"). We refer to these PEONs as PEON$_{ODD}$. To confirm the reliability of our selection criteria and ensure that the identified neurons were not specific to one trial subset, we applied the same classification procedure to the EVEN trials, identifying a separate group of PEONs referred to as PEON$_{EVEN}$. For additional comparison, maximizing statistical power and evaluating probability encoding without trial-based data splitting, we identified PEON$_{ALL}$, a group classified using all available trials.

Out of a total of 990 neurons, we identified 123 (13%) PEON$_{ODD}$, 60 neurons (6%) showed correlation with probability but no omission response, and 807 neurons (82%) showed no correlation (Fig 2C, top). On the other hand, when using the EVEN trials, 145 neurons were classified as PEON$_{EVEN}$, 69 showed correlation but no response, and 776 showed no correlation. Although the precise numbers varied slightly between subsets, the classification proportions were broadly similar, supporting the robustness of the procedure. To assess consistency more directly, we quantified the overlap between the two independently identified PEON subsets. A total of 61 neurons were classified as PEONs in both ODD and EVEN trial sets, significantly exceeding the overlap expected by chance (expected: 18 neurons; hypergeometric test, $p \ll 0.001$). This result supports the stability of the classification approach, even when applied to trial-split subsets with reduced statistical power. For comparison, when using all trials without splitting, 200 neurons were classified as PEON$_{ALL}$, 70 showed correlation but no response, and 720 showed no correlation. Pie charts for both PEON$_{EVEN}$ and PEON$_{ALL}$ classifications are shown in S2 Fig.

Fig 2D presents a box plot that aggregates the omission response of all PEON$_{ODD}$ on EVEN trials. To quantify the relationship between omission response and $O_P$ tone probability, we performed a direct Spearman's correlation analysis using all individual neuron data points, which revealed a significant positive relationship (Spearman's $\rho = 0.34$, $p = 1.6 \times 10^{-27}$, $n = 984$, representing 123 neurons × 8 probability conditions). The strength of this relationship indicates that neural responses systematically increase as the probability of the $O_P$ tone increases, providing strong evidence that these neurons encode stimulus probability.

This significant relationship confirmed the robustness of the probability-encoding property, which is likewise replicated in the analysis of PEON$_{EVEN}$ on ODD trials and PEON$_{ALL}$ using all trials (see S2 Fig). From here onward, we present the results from the PEON$_{ODD}$ group. Equivalent analyses for PEON$_{EVEN}$ and PEON$_{ALL}$ are provided in the S2–S5 Figs to further support the robustness and generalizability of our findings.

## Spatial distribution of PEONs in auditory cortex

To investigate the anatomical basis of predictive coding in the auditory system, we analyzed the laminar and areal distribution of PEON$_{ODD}$. We used current source density (CSD) analysis of local field potential (LFP) responses to determine cortical layer depths and categorized neurons into layers based on established criteria (see "Methods"). Fig 2E illustrates

the distribution of these neurons across cortical depths, demonstrating that PEON$_{ODD}$ are distributed across all cortical layers, with a higher proportion found in the supragranular (16.2%) and granular (13.4%) layers compared to the infragranular layers (9.8%). A similar distribution was also discovered from PEON$_{EVEN}$ and PEON$_{ALL}$ (see S3 Fig).

To assess whether these distributions differed significantly from a random distribution, we performed a two-sided bootstrap analysis with 300,000 samples. The analysis revealed a marginal trend in the supragranular layer ($p = 0.058$) and a significant difference in the infragranular layer ($p = 0.029$), whereas the granular layer did not show a significant deviation ($p = 0.62$). Further examination of these results indicated that PEONs were present at a lower-than-expected rate in the infragranular layer. We similarly examined neuron distribution across different auditory fields, summarized in Fig 2F, using bootstrap analysis with 300,000 samples. This analysis revealed a significantly higher proportion of PEONs in A1 ($p = 0.0036$), whereas VAF ($p = 0.9442$) and AAF ($p = 0.9926$) did not show significant deviations from the distribution expected under the null hypothesis. Further examination suggested that PEONs were present at a higher-than-expected rate specifically in A1 relative to the other auditory fields.

### Omission responses build up throughout the trials

To confirm that PEONs encode negative prediction errors, it is essential to show that their responses to omissions increase over time as predictions are gradually formed, reflecting a growing discrepancy between the predicted and actual outcomes. We analyzed PEON$_{ODD}$ from EVEN trials and focused on omissions of the O$_P$ tone in sequences where it was the standard (75%, 85%, 90%, and 95%), and therefore expected to evoke the largest omission responses. We grouped these responses by their sequence position and computed the mean firing rate ($\pm$standard error) for each bin. This grouping tests whether the omission response increases as more tones precede the omission, which is the expected pattern if these neurons are indeed signaling a negative prediction error that strengthens with increasing prediction.

Our analysis revealed a notable pattern: the magnitude of the PEONs response to these omissions progressively increased throughout the sequence, achieving a steady state or plateau after approximately 400 standards were presented (Fig 3A). The dynamics of the trial-by-trial omission response were fitted by a logistic growth curve, expressed as:

$$FR(n) = \frac{4.95}{1 + e^{-0.012*(n-223.28)}}$$

where $FR(n)$ represents the firing rate as a function of $n$, and $n$ is the bin center corresponding to the sequence position of the omissions (i.e., the number of tones presented preceding each omission). The logistic function depicts a saturating increase in response magnitude. Here, an $R^2$ value of 0.51 indicates the model's moderate success in capturing the variability in responses. To demonstrate that the observed increase in omission response was not caused by gradual changes in network activation over time, we also examined how PEONs responded to standard tones over the first 50 trials (Fig 3B). The results indicated a rapid decline in response intensity over successive trials. This decay was fitted using an exponential decay model:

$$FR(n) = 2.91 + 396.36 * e^{-2.25*n}$$

where $FR(n)$ is the firing rate at trial $n$. An $R^2$ value of 0.75, indicating a substantial fit to the data. As expected, this encapsulates the phenomenon of stimulus-specific adaptation or a decrease in positive prediction error, where the response to an expected stimulus diminishes rapidly. These trends were consistent also for PEON$_{EVEN}$ during ODD trials and PEON$_{ALL}$ over all trials (see S4 Fig).

### PEONs selectively respond to omissions while broadly reacting to tones

Finally, we investigated whether PEONs are exclusively responsive to omissions or if they also react to tone stimuli. The population responses of PEON$_{ODD}$ from EVEN trials revealed a pronounced asymmetry in omission and tone responses

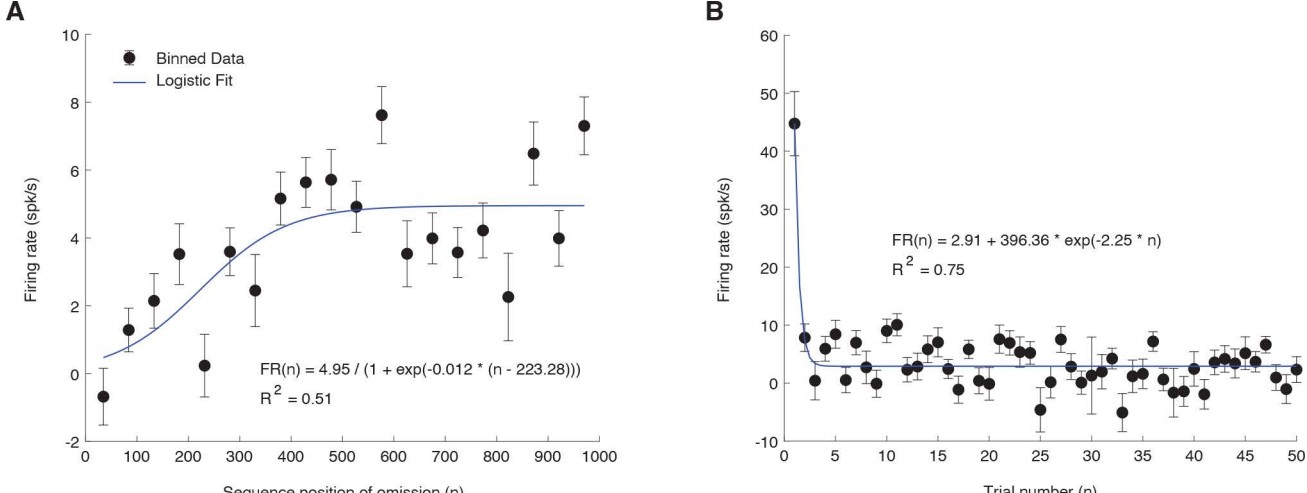

**Fig 3. Trial-by-trial dynamics of omission responses and standard tone adaptation. (A)** Firing rate of PEON$_{ODD}$ for omission events (on EVEN trials) in the four conditions where the O$_P$ tone was the standard (75%, 85%, 90%, 95%), grouped by sequence position. The x-axis represents the bin center (i.e., its sequence position), and the y-axis represents the firing rate in spikes per second. The black dots indicate the mean firing rate across all PEON$_{ODD}$ for each bin, and the error bars represent the standard error of the mean (SEM), showing variability across neurons. The blue line represents the logistic growth curve fit to the data, illustrating how omission responses evolve over successive standards. Data were averaged across sequences where the O$_P$ tone was the standard (95%, 90%, 85%, and 75%). **(B)** Firing rate of PEON$_{ODD}$ in response to the presentation of tones across the first 50 trials. The x-axis represents the trial number, and the y-axis represents the firing rate in spikes per second. The black dots indicate the mean firing rate across all PEON$_{ODD}$ for each trial, and the error bars represent SEM, showing variability across neurons. The blue line represents the exponential decay model's fit to the data, indicating a rapid decline in response intensity over the initial trials as the neurons adapt to the repeated tone presentations.

(Fig 4A, top). Omission responses showed tone selectivity based on probability: as described before for PEON$_{ODD}$, the response strength systematically increased with the probability of the O$_P$ tone. In contrast, tone-evoked responses for both O$_P$ and O$_{NP}$ tones exhibited an inverse relationship with probability. Specifically, the response to the O$_P$ tone (blue line) decreased as the probability of the O$_P$ tone increased (Spearman's $\rho = -0.19$, $p = 2.1 \times 10^{-8}$, $n = 861$, representing 123 neurons × 7 probability conditions), showing adaptation to the increasingly standard O$_P$ tone (Fig 4A, bottom). Similarly, the response to the O$_{NP}$ tone (orange line) increased as the probability of the O$_P$ tone increased (Spearman's $\rho = 0.20$, $p = 6.7 \times 10^{-9}$, $n = 861$, representing 123 neurons × 7 probability conditions), which means it also shows an inverse relationship with probability, but with respect to the O$_{NP}$ tone's own decreasing probability, reflecting deviant detection of the increasingly rare O$_{NP}$ tone. In essence, PEONs selectively signal negative prediction errors only for O$_P$ tone omissions, while their responses to both tones exhibit symmetrical adaptation to the standard tone and deviant detection of the deviant tone, without tone-specific selectivity in the direction of probability modulation.

When examining individual neurons, we found that, as expected from the definition of PEONs, their omission responses were highly selective. A substantial proportion of PEONs detected the absence of one tone while responding to both. This is illustrated in Fig 4B, which shows the responses on EVEN trials of an example PEON$_{ODD}$. This neuron's omission response (left column) increased with O$_P$ probability, exhibiting robust firing at high probabilities (90%–95%). Despite its selective omission response, it showed significant responses to both O$_P$ tone (center column) and O$_{NP}$ tone (right column), especially when the tone probabilities were low (5%–10%).

To ensure that the selectivity observed in omission responses is not merely due to residual effects from the previous trial's tone responses—such as lingering activity or altered baseline activity—we verified that omission selectivity is independent of tone selectivity. To do this, we measured the omission selectivity index (OSI) and the tone selectivity index (TSI) for each PEON (see details in "Methods"). The results showed no significant correlation between OSI and TSI across the

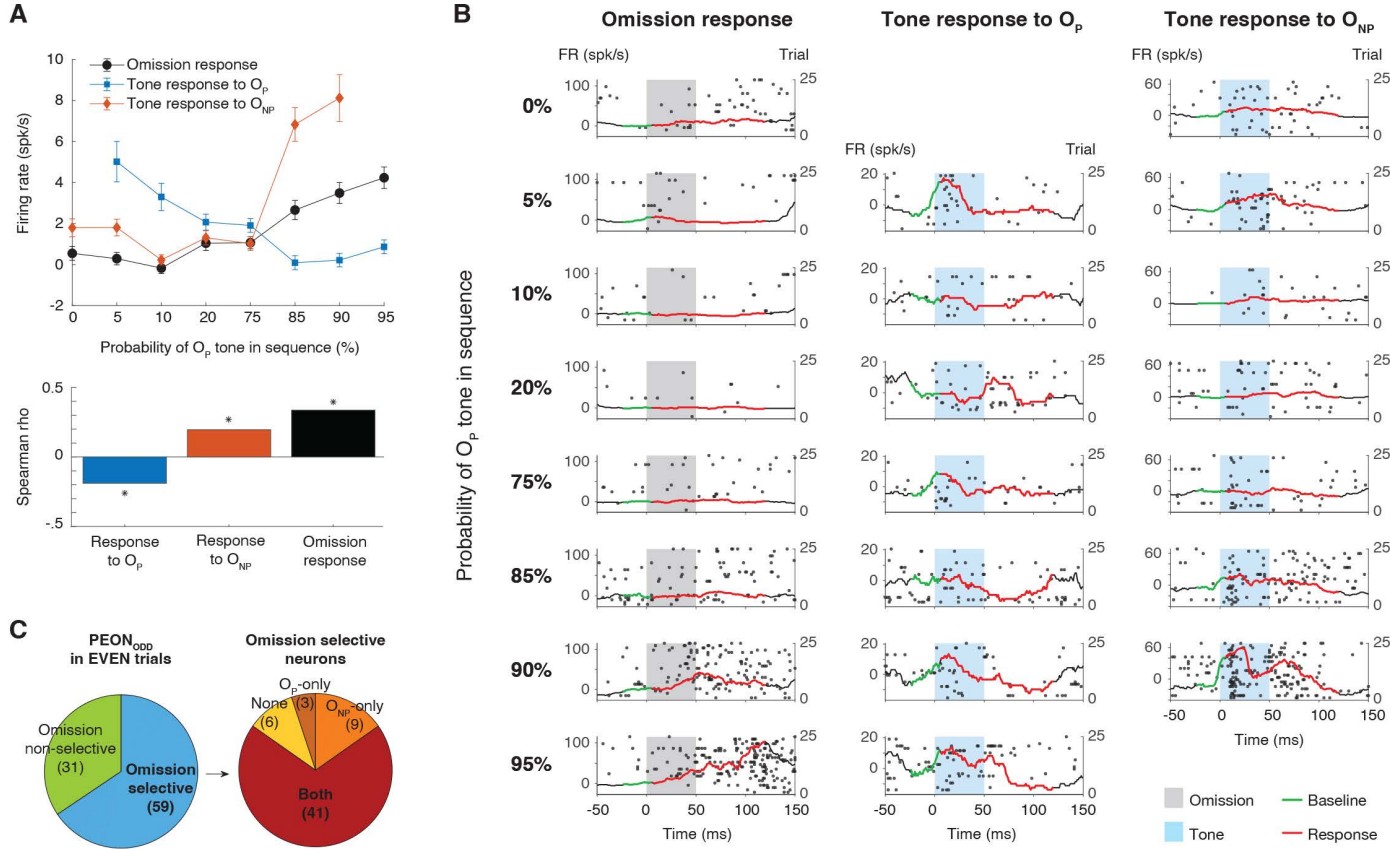

**Fig 4. Selective omission and broad tone responses in PEONs. (A)** Top: Population-level firing rates comparing omission responses (black), tone responses to the $O_P$ tone (blue), and tone responses to the $O_{NP}$ tone (red), plotted against the probability of the $O_P$ tone. The y-axis indicates firing rate (spikes/s), and error bars show standard error of the mean. All data are drawn from the EVEN trials of $PEON_{ODD}$. Bottom: Spearman's correlation coefficients between neural responses and tone probability. Bars show the correlation strength and direction for responses to the $O_P$ tone (blue), responses to the $O_{NP}$ tone (red), and omission responses (black). Asterisks indicate statistical significance (*$p < 0.05$). Underlying data are provided in S2 Data. **(B)** Raster plots (dots) and PSTHs (black lines) with omission and tone responses (red line) and baselines (green line). from a single PEON (identified in ODD trials, shown here on 25 EVEN trials), illustrating omission responses (left column), tone responses to the $O_P$ tone (middle), and tone responses to the $O_{NP}$ tone (right). Gray shading marks the omission period (expected stimulus timing), and light blue shading indicates tone presentation. For tone responses (middle and right columns), trials were subsampled to match the statistical power of omission trials, ensuring equalized trial counts across conditions. PSTHs are shown after subtracting the mean baseline (green line), and the red line highlights the evoked response. Each row corresponds to a different probability of the $O_P$ tone (0%–95%). **(C)** The left chart categorizes the 90 PEONs validated on EVEN trials, meaning they were originally identified in ODD trials and continued to show significant omission responses to $O_P$ in EVEN trials (90 of 123 $PEON_{ODD}$). These neurons were divided into selective PEONs, which responded significantly to omissions only when $O_P$ was the standard, and non-selective PEONs, which exhibited significant omission responses when either $O_P$ or $O_{NP}$ was the standard. The right chart further classifies the 59 $O_P$ omission-selective PEONs based on their tone responses—those responding to both tones, only to the $O_P$ tone, only to the $O_{NP}$ tone, or to neither.

PEON population (Spearman's ρ = −0.03, $p = 0.78$, $n = 123$ neurons). This lack of correlation indicates that a PEON's preference for a specific tone (e.g., Tone A over Tone B) does not predict its preference for the omission of that tone. Therefore, omission selectivity was not a residual effect from the previous trials.

To quantify this selectivity across the population, we examined how many PEONs exhibited significant omission responses under different conditions. Out of 123 $PEON_{ODD}$, 90 demonstrated significant $O_P$-omission responses in EVEN trials (with an additional 5 showing significant responses only to $O_{NP}$ omissions). Of these 90 neurons with $O_P$ omission responses, 59 (66%) were selective for $O_P$ omissions, whereas 31 (34%) were non-selective, exhibiting significant omission responses for both tones (Fig 4C, left). To further characterize the omission-selective group, we examined their

responses to tone presentations to determine whether they responded exclusively to one tone or to both ([Fig 4C](), right), 41 (or 69%) exhibited significant responses to both the $O_P$ and $O_{NP}$ tones across experimental conditions (see details in "[Methods]()"). Additionally, 3 PEONs responded exclusively to the $O_P$ tone, 9 solely to the $O_{NP}$ tone, and 6 did not significantly respond to either tone.

Similar trends in tone responses were observed when validating $PEON_{EVEN}$ validated on ODD trials and in $PEON_{ALL}$ (see [S5 Fig]()). In both cases, most omission-selective neurons responded to both tones, while smaller subsets responded exclusively to one tone or to neither. This consistency across datasets further supports the robustness of the omission response selectivity and its relationship to tone-evoked activity. In summary, these characteristic responses—observed in a considerable subpopulation of individual PEONs, which are selective for omissions but broadly responsive to tones— mirror the population responses and indicate an intriguing functional asymmetry. This raises the question of how computational principles and circuitry give rise to this phenomenon.

### Neural circuit model for asymmetric predictive-coding processing

To further understand how PEONs, which we presume to be negative prediction-error neurons, can display this observed asymmetry in top–down and bottom–up processing, we developed a neural circuit model consisting of multiple streams within a "predictive-coding module" ([Fig 5A]()). This model was based on the framework proposed by Keller and Mrsic-Flogel, which describes a circuit module that integrates negative prediction error computation [6].

Each predictive-coding module includes six leaky integrate-and-fire (LIF) neurons divided into two streams: one stream calculates the positive prediction error and upregulates the prediction, while the other stream calculates the negative prediction error and downregulates the prediction. A sensory neuron (denoted as I) receives the sensory signal and propagates the information in a bottom–up manner. In contrast, a prediction neuron (P) receives the prediction signal and propagates the information in a top–down manner. A positive prediction-error neuron (PE+) receives excitatory synaptic input from the sensory neuron and is suppressed by the prediction neuron via an inhibitory interneuron (I+). Conversely, a negative prediction-error neuron (PE–) receives excitatory synaptic input from the prediction neuron and is suppressed by the sensory neuron via an inhibitory interneuron (I–). All synaptic connections are static synapses that include three variables: synaptic weight, time constant, and synaptic delay (see details in "[Methods]()"). The activity of PE+ enhances the prediction signal through excitatory analog feedback, while the activity of PE– suppresses the prediction signal via inhibitory analog feedback.

In our simulation, the sensory signal was designed to replicate the experiment, featuring an analog pulse stimulus with an intensity of *S_strength* (ranging between 0 and 1) and a duration of 20 ms, delivered with a stimulus onset asynchrony (SOA) of 150 ms (see [Fig 5B](), top). Conversely, the prediction signal was modeled as a similar analog pulse as the sensory signal, but varied in intensity (*P_strength*), which could be enhanced by excitatory feedback from PE+ or diminished by inhibitory feedback from PE– ([Fig 5B](), bottom). It is important to recognize that our model assumes the temporal profile of the prediction signal is learned through a separate mechanism, while the predictive-coding module adjusts only the signal's magnitude based on prediction error.

Predictive coding modules from different streams, each processing a specific frequency according to a tonotopic organization, are interconnected through lateral excitatory synaptic connections. These connections run from PE+ of one module to PE– of another (as illustrated between streams A and B in [Fig 5C]()). Intuitively, this setup enables PEONs, represented as PE–, in one stream to respond to stimuli in another stream, endowing them with a broader bottom–up receptive field. Moreover, this configuration allows an unexpected stimulus in one stream to inhibit the predictive signal in other streams. For example, as shown in [Fig 5C](), if the sensory signal in Stream B is unexpected, PE+ in Stream B will activate, causing PE– in Stream A to also activate through the lateral connection. This activation will then suppress the predictive signal in Stream A. Thus, when an unexpected Tone B is detected, indicating that Tone A (or other tones) is unlikely, the prediction for Tone A (or other tones) is consequently reduced. Conversely, since the absence of an expected Tone B does

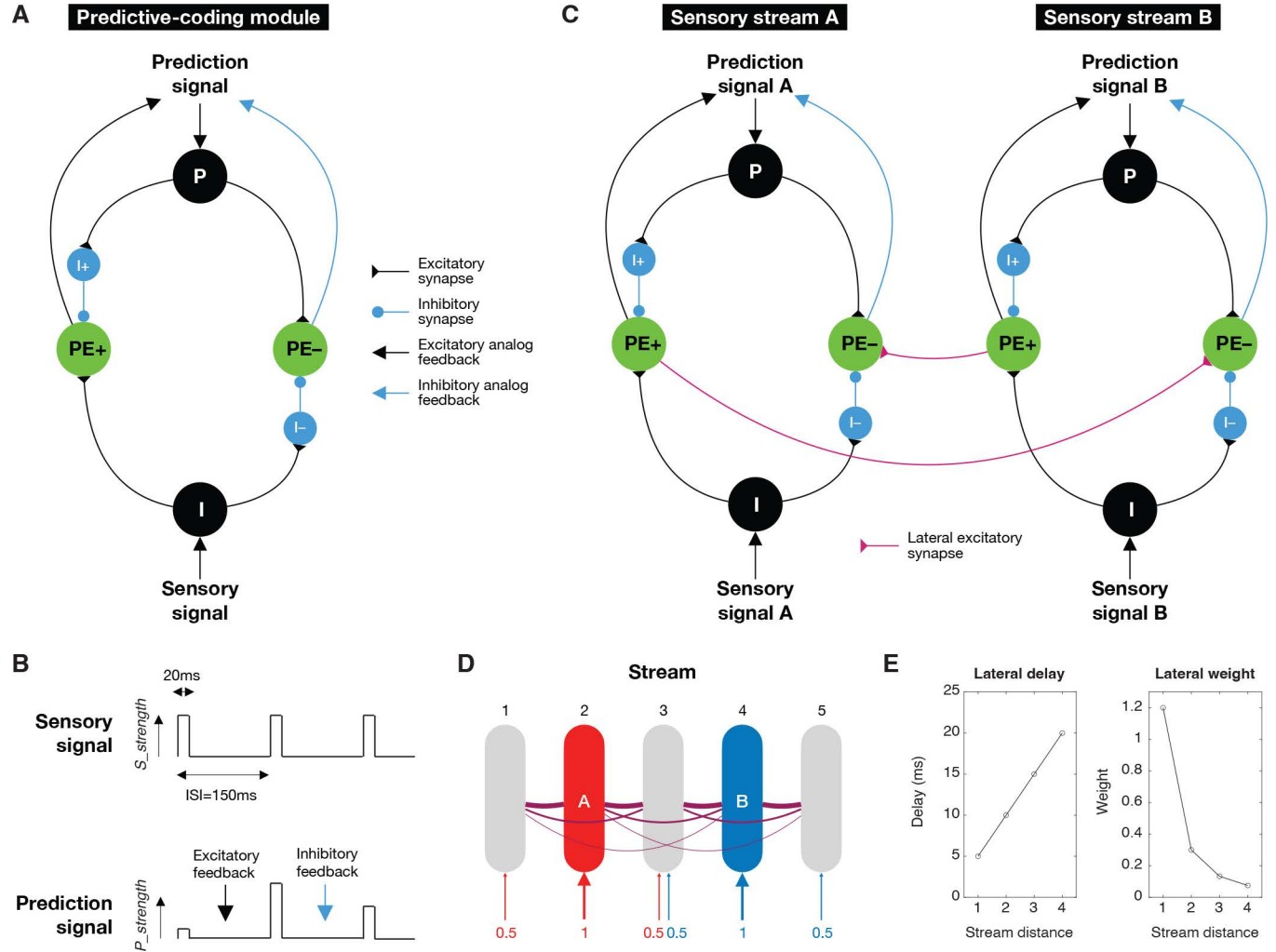

**Fig 5. Neural circuit model for asymmetric predictive-coding processing. (A)** Basic circuit module showing interconnections between neurons. P: prediction neuron, PE+: positive prediction error neuron, PE−: negative prediction error neuron, I+/I−: inhibitory interneurons, I: sensory input neuron. The connection types are also indicated. **(B)** Temporal profiles of sensory and prediction signals. **(C)** Lateral connections between two example circuit modules processing different sensory streams. **(D)** Full model with 5 sensory streams showing tonotopic organization. Streams 2 and 4 receive full strength input for tones **A** and **B**, respectively. The numbers below illustrate the strength of sensory signal (*S_strength*) in each stream. **(E)** Distance-dependent properties of lateral connections, showing delay and weight as a function of the distance between streams (see more details in "Methods").

not necessarily suggest the presence of Tone A (or other tones), our model does not include lateral connections running from PE− to PE+.

Our comprehensive model includes five streams: Stream 2 is tuned to detect the $O_P$ tone (Tone A), while Stream 4 detects the $O_{NP}$ tone (Tone B) (see Fig 5D). The model incorporates a simple receptive field structure for sensory signals. Specifically, Stream 2 receives Tone A at full strength (*S_strength* = 1), whereas the adjacent Streams 1 and 3 receive Tone A at half strength (*S_strength* = 0.5). Similarly, Stream 4 receives Tone B at full strength, and the adjacent Streams 3 and 5 receive Tone B at half strength. Additionally, the model features lateral excitatory connections between streams that are distance-specific (Fig 5E). The synaptic delay between any two streams is proportional to their spatial separation, and the synaptic weight decreases with the inverse square of their distance (see more details in "Methods").

## Asymmetric activities in modelled PEON

We conducted multiple simulations using the model, each featuring unique probability combinations of tones A and B, along with a consistent 5% omission rate, mirroring the conditions of our rodent experiments. Each simulation included 500 stimuli—either tone A, B, or an omission (O)—across a total of 75 s. Importantly, every simulation began with no initial predictions across all streams ($P\_strength = 0$).

Here, we demonstrate the model's responses using the example where the sensory inputs consist of 90% A, 5% B, and 5% O, randomly ordered as shown in Fig 6A. The spiking activities of Neurons I, P, PE+, and PE− across the five streams are illustrated in Fig 6B. Notably, Neuron P began firing after a few seconds, particularly in Streams 1–3, which are responsive to Tone A. This activity indicated that the prediction for Tone A was beginning to be established. Consequently,

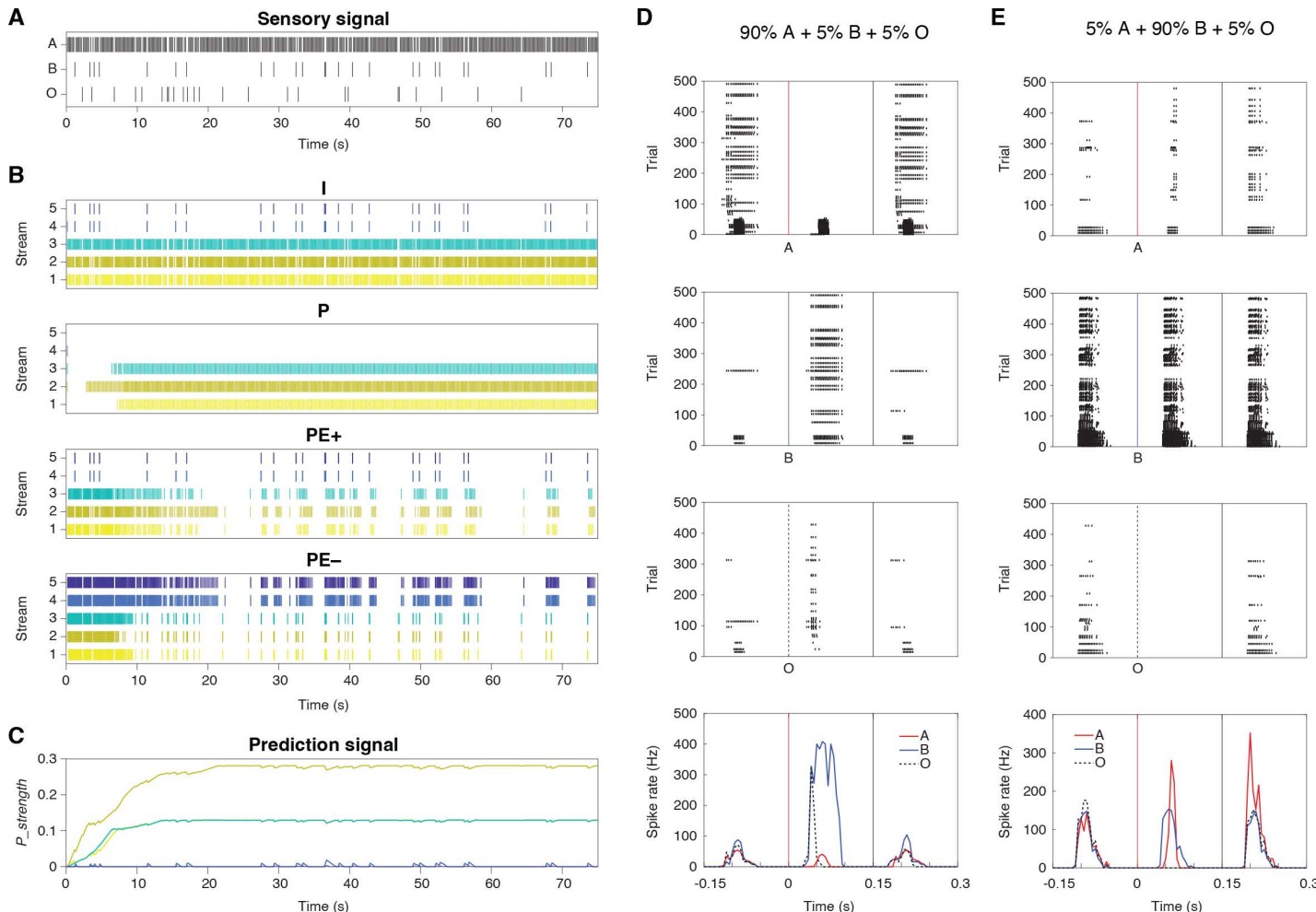

**Fig 6. Asymmetric activities in modelled PEON. (A)** Sensory input sequences over 75 s, showing occurrences of tones **A** (90%), **B** (5%), and omissions (5%). **(B)** Spiking activities of neurons I, P, PE+, and PE− across five streams. Colors represent different streams. **(C)** Prediction signal strength (*P\_strength*) over time for different streams. The same color scheme is used as in panel B. **(D)** Raster plots (top 3 panels) and PSTHs (bottom panel) for PEON in Stream 2 under: 90% A, 5% B, 5% O. Responses to tones A (red vertical line), B (blue vertical line), and omissions (O, black dotted vertical line) are shown separately. PSTHs display firing rate over time, with colored lines indicating responses to different stimuli. In the top three panels, spike rasters are plotted across the same set of 500 trials. Because Tone B occurs in only 5% of trials, fewer spikes are displayed for **B** compared to A. The bottom panel shows the corresponding firing rates for each condition. **(E)** Raster plots and PSTHs for PEON in Stream 2 under: 5% A, 90% B, 5% O.

the positive prediction error, reflected by PE+'s spiking activity, began to diminish. Additionally, the spiking activity of PE−, primarily associated with the omission of Tone A, was also observed in Streams 4 and 5. This was largely due to lateral connections from PE+ in Streams 1–3. The prediction signals across the five streams, detailed in Fig 6C, show that *P_ strength* increased and plateaued in Streams 1–3, caused by PE+'s firing, with Stream 2 displaying the highest value.

To explore asymmetric predictive-coding processing, we focused on the PEON in Stream 2, the target neuron for modeling our experimental results. Our goal was to reproduce the experimentally observed asymmetry: selective responses to omissions of the preferred tone, but broad responses to both tones. The raster plots of this PEON, presented in Fig 6D for a simulation containing 90% A, 5% B, and 5% O, characterize responses to different stimuli: Tones A, B, and omissions O. The PSTH reveals that while the PEON consistently responded to omissions, it did not respond to Tone A (in later trials), which was anticipated. Importantly, the PEON did respond to Tone B, a response enabled by lateral connections. In Fig 6E, we present raster plots from a different simulation with 5% A, 90% B, and 5% O. Similar to the previous simulation, the PEON in Stream 2 responded to Tone B via lateral connections. It also reacted to Tone A, facilitated by the lateral connections from nearby Streams 1 and 3, where PE+ was activated by the deviant Tone A. Notably, there was no response to omissions, as the prediction in Stream A was not effectively established due to the minimal occurrence of only 5% A. In conclusion, the PEON in Stream 2 demonstrated the asymmetry of receptive fields in bottom–up sensory and top–down predictive processing, reflecting the findings in the rodent experiment.

Note that this asymmetry requires that the weights of the lateral excitatory connections from nearby neurons (as shown in Fig 5E) exceed the inhibitory weights within the module (see S1 Table). This setup ensures that sensory inputs from the neighboring streams can overcome inhibition to elicit a response on PEONs. When the number of lateral neighbor neurons increases (there are only two in our model), reduced excitatory weights can remain effective. Furthermore, we believe these lateral connections could be activity-dependent (e.g., via spike-timing-dependent plasticity) and further refined through self-organization. However, this lies beyond the scope of our current study.

## Alternative models and evaluation

To assess the validity of the proposed model with lateral connections (the model labeled "Lateral PE+ to PE−"), we evaluated two alternative models and compared their predictions against the experimental data. The alternatives included: a model with lateral connections from Neuron I to PE− across different streams (termed "Lateral I to PE−", depicted in Fig 7A), and a model with no lateral connections at all (referred to as "No lateral"). For each model, we measured the responses of the PEON in Stream 2 to Tones A, B, and omission O under different probabilities of Tone A (Prob(A)): from 0% (95% B and 5%O) to 95% (0% B and 5%O). The results are shown in Fig 7B, together with the correlation between the firing rate and Prob(A).

In our proposed model, the PEON's response to omission significantly correlated with Prob(A) (Spearman's $\rho = 0.79$, $p = 2.1 \times 10^{-108}$, $n = 500$ trials; see "Methods" for further details), indicating that the PEON did encode the probability of the stimulus within its stream. For Tone A, the PEON exhibited reduced responses as Prob(A) increased, demonstrating a significant negative correlation between firing rate and Prob(A) ($r = -0.73$, $p < 1 \times 10^{-250}$, $n = 4,750$ trials). This occurred because PE+ in nearby Streams 1 and 3 detected less surprise at higher Prob(A) values, leading to less activation of the PEON in Stream 2 through lateral connections (as also noted in Fig 5E). Conversely, for Tone B, the PEON's response positively correlated with Prob(A) ($r = 0.81$, $p < 1 \times 10^{-250}$, $n = 4,750$ trials). At higher Prob(A) values, Tone B emerged as the deviant tone. This led to an enhanced detection of surprise by PE+ near Stream 4 for Tone B, which subsequently activated the PEON in Stream 2 more strongly through lateral connections.

In the alternative "Lateral I to PE−" model, the PEON's response to omission was also correlated with Prob(A) ($r = 0.82$, $p = 4.3 \times 10^{-123}$, $n = 500$ trials). For Tone A, unlike the proposed model, the PEON in Stream 2 did not exhibit decreased responses at high Prob(A) values. This occurred because the PEON directly received sensory input from Neuron I, which remained unchanged across different Prob(A) settings. Despite this, a positive correlation between the firing rate and

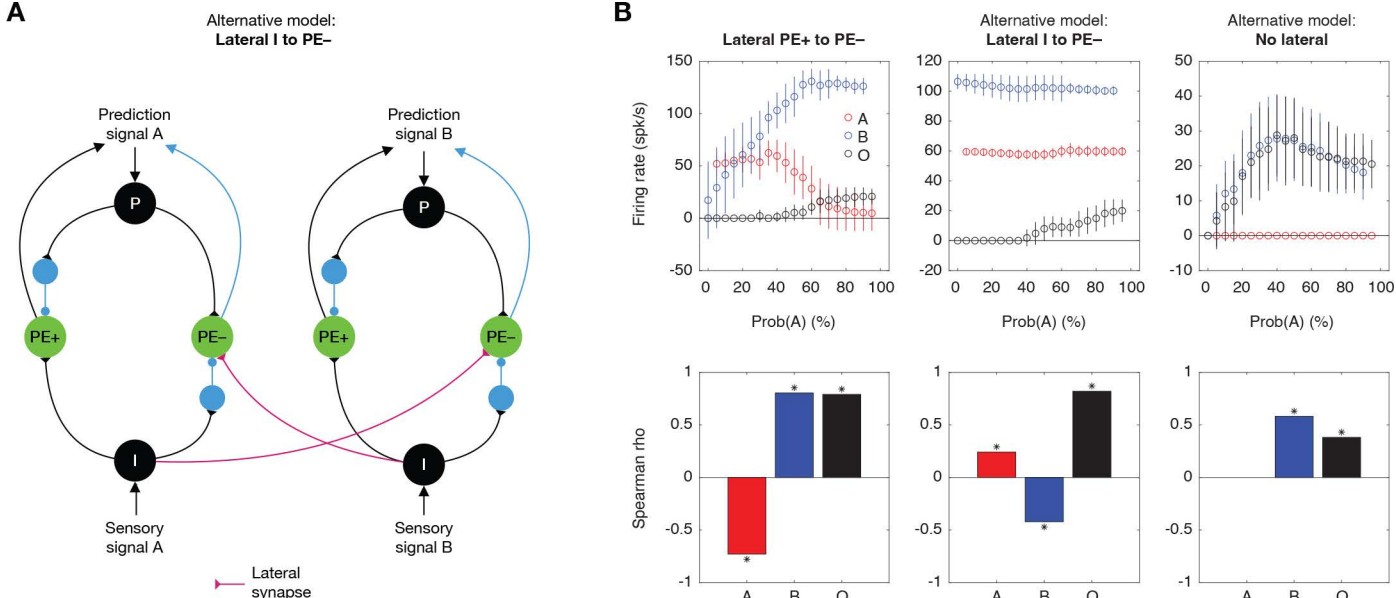

**Fig 7. Alternative models and evaluation. (A)** Schematic of the alternative model with lateral connections from input neurons (I) to negative prediction error neurons (PE−) across streams **A** and B. **(B)** Evaluation of alternative models: Lateral PE+ to PE−, Lateral I to PE−, and No Lateral. Top panels show the firing rates of PEON in Stream 2 in response to Tone A (red), Tone B (blue), and omissions (O, gray) across varying probabilities of Tone A (Prob(A)). Bottom panels display correlations between firing rates and probabilities for Tones A, B, and omissions for each model.

Prob(A) was observed ($r = 0.82$, $p = 4.3 \times 10^{-123}$, $n = 4{,}750$ trials). This was due to slightly increased responses from Neuron I in nearby Streams 1 and 3, which were caused by an elevated membrane potential resulting from frequent stimulation by Tone A. Conversely, for Tone B, the opposite effect occurred: Neuron I near Stream 4 showed an increase in membrane potential due to frequent stimulation of Tone B at low Prob(A) levels. Therefore, despite only minor changes in firing rate across different Prob(A) values, a negative correlation was observed ($r = -0.42$, $p = 3.5 \times 10^{-204}$, $n = 4{,}750$ trials).

In the alternative "No lateral" model, the PEON's response to omission correlated with Prob(A) ($r = 0.38$, $p = 9.2e^{-19}$, $n = 500$ trials), as this encoding occurs within the stream without the need for lateral connections. For Tone A, the PEON displayed no responses, which was anticipated since it could only react to an omission in the absence of lateral connections. For Tone B, the PEON exhibited similar responses as to omissions, since in Stream 2, any stimulation differing from Tone A was treated as an omission. Consequently, a similar positive correlation was observed ($r = 0.58$, $p < 1 \times 10^{-250}$, $n = 4{,}750$ trials).

To validate these models, each demonstrating unique encoding characteristics, we compared them to data from our rodent experiment. In our analysis, Tones A and B corresponded to the $O_P$ and $O_{NP}$ Tones, respectively. The correlation patterns we observed in our experimental data (see Fig 4A) align closely with the predictions of our proposed model. The positive correlation for omission responses, negative correlation for responses to $O_P$ tone, and positive correlation for responses to $O_{NP}$ tone are all successfully captured by the model with lateral PE+ to PE− connections. These results support the proposed model with its lateral connections facilitating positive prediction errors, distinguishing it from the alternative models.

The models discussed so far have not taken into account that the strength of sensory signals could diminish due to frequent exposure to a specific stimulus. To address this, we analyzed responses from neurons in the thalamus ($n = 130$ neurons) across various probabilities of the $O_P$ Tone (see S6A Fig and "Methods" for details). Accordingly, we modeled the sensory signal strength, $S\_strength$, as a function of Prob(A) (S6B Fig) and conducted simulations. By simulating various

levels of sensory adaptation (see "Methods"), our results reaffirmed that only the proposed model successfully replicated the probability encoding characteristics observed in the rodent data (S6C Fig).

## Computational benefits of lateral connections

Our next step was to explore the potential computational benefits of the proposed lateral connections, particularly in terms of enhancing prediction encoding. How do these connections influence prediction signals? Without lateral connections, not only would Stream 2 predict Tone A, but also the neighboring streams (e.g., Stream 1), due to their broad sensory receptive fields receiving half-strength sensory signals. However, with lateral connections, these predictions are reduced not only by the omission of Tone A but also by the presence of Tone B. Particularly, we theorized that this additional reduction would suppress the already low prediction signals in the neighboring streams. This leads to a scenario where the prediction signal is predominantly strong in Stream 2, thus resulting in a more precise encoding of Prob(A) across streams.

To test this, we first assessed the steady value of *P_strength* after the prediction signal stabilized across various stimulation sequences (as in Fig 6C). These sequences varied Prob(A) from 0% to 95%, with increments of 1%. In Fig 8A, we plotted the time courses of *P_strength* for each 75-s simulation under different Prob(A) settings, focusing on Stream 2, where our target PEON was situated, and the adjacent Stream 1. For comparison, the time courses for Streams 1 and 2 in the model without lateral connections ("No lateral") are shown in Fig 8B. The steady value of *P_strength* was determined

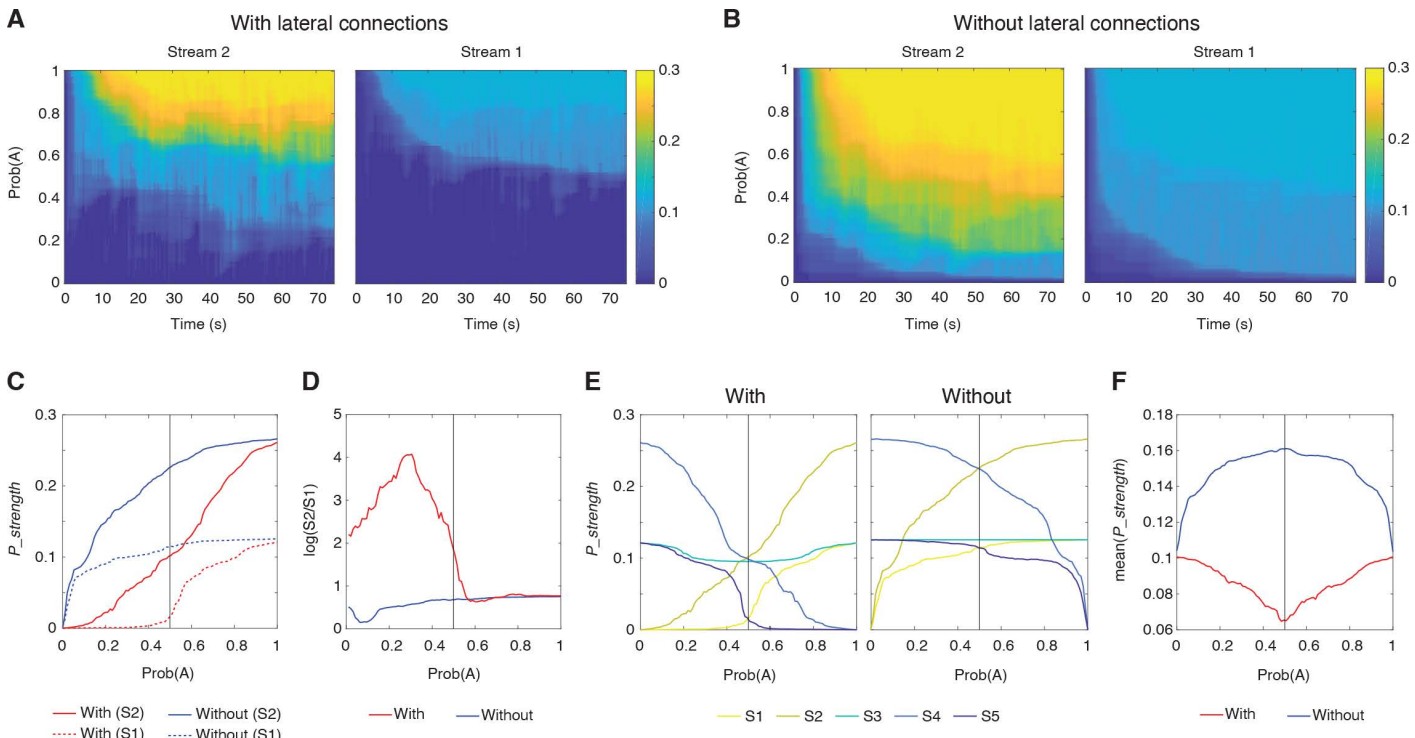

**Fig 8. Computational benefits of lateral connections. (A)** Time courses of prediction strength (*P_strength*) in Streams 2 and 1 with the lateral connections for varying probabilities of Tone A. Colors indicate the magnitude of *P_strength*. **(B)** Time courses of prediction strength in Streams 2 and 1 without the lateral connection. The same representation is used as in the panel A. **(C)** Steady-state *P_strength* in Stream 2 (solid lines) and Stream 1 (dotted lines) as a function of Prob(A) with (red) and without (blue) lateral connections. **(D)** Contrast in *P_strength* between Streams 1 and 2 (log scale) with and without lateral connections, plotted against Prob(A). **(E)** Steady-state *P_strength* across all streams (S1 to S5) as a function of Prob(A), with and without lateral connections. **(F)** Mean *P_strength* across all streams for different Prob(A) settings, comparing the network with (blue) and without (red) lateral connections.

by averaging values over the last 50 s (from 25 to 75 s). It was plotted as a function of Prob(A) for both models, with and without lateral connections, across Streams 1 and 2 (as shown in Fig 8C). In both streams, *P_strength* increased as Prob(A) rose. However, the increase was less pronounced in the model with lateral connections, especially at lower Prob(A) values. We further analyzed the contrast in *P_strength* between Streams 1 and 2 by calculating the decibel value (logarithmic ratio of *P_strength* in Stream 2 to that in Stream 1). This contrast was significantly greater in the model with lateral connections compared to the model without, particularly when Prob(A) was less than 0.5 (Fig 8D). This suggested that the lateral connections enhance the contrast of prediction signals across streams.

We also hypothesized that the additional suppression of prediction signals facilitated by lateral connections could enhance prediction encoding efficiency. This is supported by the observation that *P_strength* was generally lower with lateral connections (as shown in Fig 8C). To investigate this further, we measured *P_strength* across Prob(A) for all streams in both models (Fig 8E) and then calculated the average *P_strength* across streams for each Prob(A) setting (Fig 8F). The findings suggest that with lateral connections, the network requires less "energy" or smaller prediction signals to effectively encode sequences, particularly in less predictable scenarios where the occurrences of Tones A and B are equal, such as Prob(A) = 0.5. This demonstrates a more efficient encoding strategy. Conversely, without lateral connections, the network expends more energy to manage more unpredictable sequences.

## Discussion

This study aimed to understand how negative prediction-error neurons in the rat auditory cortex respond. We combined electrophysiological recordings with computational modeling to demonstrate an asymmetry in prediction-error signaling, which we attribute to "lateral prediction suppression" where positive and negative prediction errors interact through lateral connections to enhance the precision and efficiency of prediction encoding.

### Auditory omission responses as correlates of negative prediction error

Omission paradigms are a powerful tool for isolating negative prediction-error signals, as they eliminate the confounding influence of sensory input present in positive prediction error paradigms (e.g., oddball sequences). In human studies using EEG and MEG, mismatch negativity (MMN) responses to unexpected omissions in tone sequences have been reported, which are interpreted as evidence for predictive coding mechanisms [17,20,22–24,33]. More recently, single-neuron recordings in animals have begun to reveal the neural mechanisms underlying these auditory prediction errors. Studies have reported omission-responsive neurons in the auditory cortex of both awake and anesthetized rodents. One study found that 35.6% of neurons in awake animals and 19.5% of neurons in anesthetized rats and mice exhibited significant responses to the omission of expected tones [34,35]. Another study, using more complex stimuli involving the omission of expected gaps within auditory sequences, also demonstrated robust omission responses in unanesthetized rats [25]. Alongside these auditory findings, multiple studies in mouse visual cortex have documented sensorimotor prediction-error signals, specifically negative prediction errors, in response to unexpected reductions in expected sensory input [7,36,37]. These studies demonstrate that neurons in V1 exhibit negative mismatch responses when expected visual flow is weakened or omitted during locomotion. A molecularly distinct population of neurons in V1 has been identified that selectively responds to unexpected omissions of predicted visual stimuli [9] Extending this sensorimotor framework to the auditory system, a recent study showed that neurons in the auditory cortex responded to unexpected reductions in movement-contingent sound within a virtual reality environment, signaling a negative prediction error [28]. Building upon this foundation, our study's conceptual advance stems from our paradigm design that systematically varies tone probabilities while maintaining consistent omission rates, allowing us to characterize negative prediction error as a continuous, probability-encoding mechanism rather than simply a binary deviance detector. This approach also reveals the dynamic nature of these prediction signals, which develop gradually through repeated exposure to statistical patterns in the environment. The subsequent sections will explore how these findings, along with our observations

of asymmetry in predictive processing, led us to develop a circuit-level model that offers a new perspective on information flow in predictive coding networks.

## Top–down "predictive field"

Our model builds upon the framework proposed by Keller and Mrsic-Flogel, which describes predictive coding as a canonical cortical computation [6]. Their model emphasizes hierarchical processing with distinct neuron populations computing positive and negative prediction errors. Consistent with this framework, our findings provide evidence for both types of prediction error in the rat auditory cortex. While we focused on characterizing negative prediction errors in PEONs, we also observed oddball responses in the broader population of non-PEON neurons (S7 Fig), with enhanced responses to unexpected tones. Since such responses have been interpreted as positive prediction-error signals in previous studies, we infer that at least some of these neurons encode positive prediction error [16,18,38–41]. More importantly, our model additionally describes how predictive coding modules, which process bidirectional information, interact both vertically and horizontally across sensory streams. This provides a means for prediction errors to integrate across bottom–up receptive fields. Moreover, we hypothesize that there is also a receptive field for top–down signaling. Traditionally, a neuron's receptive field is defined by the range of external stimuli that can directly evoke its activity, a concept rooted in bottom–up sensory processing. However, neurons like PEONs are influenced not only by bottom–up sensory input but also by prediction signals generated in other brain areas. Therefore, their effective response properties – what we might term their "predictive field" – are shaped also by the specificity of the prediction. We believe that the asymmetry between the bottom–up receptive field and the top–down predictive field provides a means to integrate predictions and prediction errors, and that it could be inverted (i.e., a larger predictive field than receptive field) at other hierarchical levels.

## Lateral connections in predictive coding networks

Our circuit model proposes that unexpected stimuli in one frequency stream inhibit predictive signals in adjacent streams via lateral prediction suppression. This enhances the contrast between the predicted frequency and others, sharpening the prediction signal and facilitating more efficient updates to the internal model, conceptually analogous to lateral inhibition in sensory systems, which serves to enhance the contrast of bottom–up sensory input [42]. That is, the lateral prediction suppression we propose for the top–down predictive field is functionally analogous to lateral inhibition in the bottom–up receptive field.

Biological evidence strongly supports the existence of excitatory lateral connections between adjacent frequency channels in A1. These connections play a crucial role in spectral integration and the processing of complex auditory stimuli [43–45]. Anatomical studies have revealed that pyramidal neurons in A1 exhibit long-range horizontal collaterals that form connections across different frequency representations, often aligning with isofrequency bands [46,47]. Physiological investigations have demonstrated that these intracortical pathways contribute significantly to the breadth of subthreshold frequency receptive fields, which can span up to five octaves [48,49]. The strength and probability of these connections are distance-dependent, with stronger and more probable connections between neurons tuned to close frequencies, weakening as the frequency distance increases [50,51]. This organized network of lateral connections facilitates the integration of spectral information across different frequency channels, supporting complex auditory processing and cortical plasticity in A1.

Similar patterns of lateral connectivity have been observed in other sensory modalities, such as orientation-specific connections in visual cortex [52] and whisker-related connections in somatosensory barrel cortex [53], suggesting a common organizational principle across different sensory areas. The development of these lateral connections might involve both innate and learned components. While the basic framework of these connections may be genetically predetermined, their refinement occurs during early postnatal development through experience-dependent plasticity [54]. This developmental process allows for the fine-tuning of these connections based on sensory experience, optimizing the auditory cortex for processing the specific acoustic environment encountered during critical periods of development.

In our model, the one-way excitation from PE+ to PE− neurons allows for rapid recalibration of expectations across streams. This asymmetry enables efficient updating of predictions based on positive surprises while avoiding false alarms from mere absences. Integration across frequency channels enables complex interactions, with sequences of tones providing predictive information about subsequent tones. Naturalistic sounds like tone clouds can activate multiple streams simultaneously, leveraging the network's capacity for sophisticated auditory processing [55].

## Lateral prediction suppression and the free energy principle

Building upon these anatomical and functional observations of lateral connectivity, we can consider how our proposed circuit model might be interpreted through theoretical principles. The free energy principle [1,56] potentially offers an elegant explanatory framework for understanding why these lateral connections may have evolved. Could the lateral prediction suppression we observed be more than just a convenient circuit mechanism? When viewed through the lens of free energy minimization, these connections might represent an optimal solution for efficient predictive processing. In S1 Text, we provide a mathematical treatment that suggests when a simplified form of the free energy principle is applied to a two-tone paradigm similar to our experimental design, lateral interactions between prediction errors across different streams emerge naturally from the formalism.

In our mathematical exploration, we considered a simplified generative model where auditory tones at different frequencies are represented as hidden variables with linear relationships to sensory inputs. By deriving the update rules for prediction signals under free energy minimization, we found that the resulting equations comprise three key components: updates based on prediction errors within the same frequency stream, lateral interactions of prediction errors across different frequency streams, and a temporal decay term. Notably, the lateral interaction term emerges from the correlation structure of sensory inputs across different frequencies—a statistical regularity well-documented in natural auditory environments [57]. These theoretical derivations align remarkably well with our proposed circuit model, where lateral connections between PE+ and PE− neurons across different frequency streams play a crucial role in shaping prediction signals.

This theoretical perspective raises intriguing possibilities about why the brain might implement the asymmetric receptive fields we observed. The efficient coding strategy—where unexpected stimuli in one frequency channel not only signal prediction errors within that channel but also laterally modulate predictions in neighboring channels—could represent an evolutionarily advantageous solution for minimizing computational costs while maximizing predictive accuracy. Such a theoretical framing might also explain the directionality of lateral connections we proposed, flowing primarily from positive to negative prediction error neurons rather than bidirectionally. While experimental validation of these theoretical predictions remains a challenge for future work, the mathematical alignment between our empirical findings and free energy principles suggests that lateral prediction suppression could be a fundamental computational motif in predictive processing, potentially extending beyond the auditory system to other sensory modalities.

## Layer and area specificity of PEONs

Our finding that PEONs are more prevalent in granular and supragranular layers aligns with current views on the laminar organization of predictive coding circuits [4]. These layers are thought to be primary sites for prediction error computation and integration of top–down predictions with bottom–up sensory input. The relative scarcity of PEONs in infragranular layers suggests these neurons may be less involved in generating feedback predictions to earlier processing stages. Regarding their distribution across different auditory fields (A1, VAF, AAF), while PEONs were present in all areas examined, they showed a significantly higher proportion in A1 compared to VAF and AAF (based on bootstrap analysis). This potential enrichment in A1 might suggest functional specializations within primary auditory cortex related to negative prediction error signals, although the presence of PEONs across fields still supports the idea that predictive coding is a widespread cortical computation [6]. However, this raises questions about potential functional specializations not captured by our current paradigm.

## Dynamics of omission responses

A notable feature of the PEON omission responses is their shape; rather than exhibiting a sharp, transient peak as seen in typical tone responses, they often display a more gradual onset and prolonged duration [27,32,34] compared to both typical auditory evoked responses and the idealized, instantaneous prediction-error signals assumed in simplified predictive coding models, including our own. While our model effectively captures the probability encoding and asymmetry of PEON responses, it does not explicitly model the detailed temporal dynamics. Functionally, a more gradual negative prediction-error signal might help prevent drastic responses to slightly delayed expected input—contrasting with the rapid responses needed for unexpected stimuli. Furthermore, the negative prediction-error signal could reflect a prediction not only about the content (i.e., the tone) but also about its timing. In other words, the prediction signal could be gradually ramped up according to a hazard function, modeling how the brain adjusts its anticipation of an event as time passes [58–61], and the negative prediction-error signal could follow a similar pattern. We suspect that this ramp-up response involves a recurrent cortical circuit normally inhibited by sensory input; when the input is omitted and disinhibition occurs, the negative prediction-error signal gradually increases. Future computational modeling that explicitly incorporates recurrent connectivity and temporal integration mechanisms will be crucial for elucidating the precise neural mechanisms underlying this observed latency.

## Establishment of prediction signals

A notable limitation of our study is the absence of explicit "prediction neurons" in both our experimental findings and computational model. While we identified neurons responding to prediction errors, we did not find neurons directly representing predictions about upcoming stimuli. Predictive signals might be generated in higher-order auditory areas, non-auditory regions providing top–down input, or encoded in a distributed manner across neuronal populations. Further computational modeling of distributed prediction encoding could help guide future experiments by suggesting specific patterns or population-level dynamics to look for in neural data.

It is important to consider whether the observed omission responses might themselves be a prediction signal, rather than solely a prediction error. However, several aspects of the PEON activity argue against this interpretation. First, a prediction signal should precede the expected time of the stimulus, while PEON omission responses occur after the expected tone onset. Second, prediction signals should appear consistently across all trials, regardless of whether the stimulus is presented or omitted. Instead, we observe that PEONs exhibit markedly different activity patterns between presentation and omission trials—in some cases showing smaller responses when tones are presented compared to when they are omitted. Thus, the PEON activity pattern is more consistent with a negative prediction-error signal, reflecting the mismatch between expectation and reality, rather than just the expectation itself.

Future research should address the limitations of our study, such as the potential effects of anesthesia on neural responses, by comparing findings with those from awake and behaving animals. Additionally, employing more complex stimuli and varying spectral distances between tones could further probe the limits of the proposed lateral prediction suppression mechanism. Another important direction is identifying the specific neuronal populations involved in prediction-error signaling. While our study focused on firing rate dynamics, different neuronal subtypes may play distinct roles in encoding positive and negative prediction errors. Techniques such as optogenetic tagging, juxtacellular recording, or genetic labeling could help determine whether PEONs correspond primarily to excitatory pyramidal neurons or specific subtypes of inhibitory interneurons, providing critical insights into the circuit mechanisms underlying predictive processing in the auditory cortex.

## Methods

All experimental procedures were conducted in strict accordance with the ethical guidelines established by the Japanese Physiological Society for the care and use of animals in physiological studies. The experimental protocol was approved by the committee on the ethics of animal experiments at the Graduate School of Information Science and Technology, the University of Tokyo (JA21-9).

PLOS Biology

## Animals

The study involved 10 male Wistar rats, aged 11–12 weeks, and weighing between 250 and 350 gms. The animals were housed in a carefully controlled environment with a 12-h light/dark cycle, ensuring optimal conditions for their well-being. Food and water were provided ad libitum throughout the study. To minimize animal suffering, all surgical procedures were conducted under appropriate anesthesia. Upon completion of the experiments, the animals were humanely euthanized using an overdose of pentobarbital sodium (160 mg/kg, administered intraperitoneally), following established guidelines for the humane termination of animal studies.

## Surgical procedures

The rats were anesthetized using urethane, with an initial dose of 1.5 g/kg administered intraperitoneally and supplementary doses of 0.5 g/kg provided as needed to ensure stable and deep anesthesia. This depth of anesthesia was confirmed by the absence of corneal and pedal reflexes. The animals were secured using a custom-made head-holding apparatus. Atropine sulfate (0.1 mg/kg) was administered subcutaneously at the onset of the procedure to reduce bronchial secretion viscosity. Local anesthetic xylocaine (0.3–0.5 ml) was applied at the incision site before making the skin incision. A needle electrode was inserted subcutaneously into the right forepaw to serve as ground. A small craniotomy was performed near the bregma landmark to place a reference electrode, ensuring electrical contact with the dura. A larger craniotomy was performed over the right auditory cortex (4.0 mm posterior and 6.0 mm lateral to bregma), and the right temporal muscle, cranium, and dura covering the auditory cortex were surgically removed. The exposed cortical surface was kept moist with saline to prevent drying. Cisternal cerebrospinal fluid drainage was conducted to minimize cerebral edema. Throughout the experiment, an adequate and stable level of anesthesia was ensured by monitoring the animals' respiratory rate, heart rate, and hind-paw withdrawal reflexes.

## Electrophysiological recordings

Neuronal activity was recorded using both a custom-designed surface microelectrode array [62] and a Neuropixels 1.0 electrode array (IMEC, Belgium) [63].

## Mapping auditory cortex using the surface array

To record neuronal activity epipially over the auditory cortex, a surface array (NeuroNexus, Ann Arbor, MI, USA) comprising 64 platinum electrodes was employed. Each electrode had a diameter of 100 μm and a center-to-center distance of 500 μm, covering an area of 4.5 × 3.0 mm. The array featured 0.3-mm diameter holes between the electrodes to allow the insertion of a Neuropixels array. Auditory evoked potentials (AEPs) were recorded using amplifiers with a gain of 1,000, a digital filter bandpass of 0.3–500 Hz, and a sampling frequency of 1 kHz, employing the Cerebus Data Acquisition System (Blackrock Microsystems LLC, Salt Lake City, UT, USA).

We mapped the click-evoked responses to pinpoint the location of the auditory cortex (AC). A click was defined as a monophasic positive wave lasting 20 ms. 60 clicks were administered at a rate of 1 Hz, and LFPs were recorded. The peak amplitude of the middle-latency auditory-evoked potential (P1) within the first 50 ms following the stimulus onset was measured from the grand averaged LFP. To identify three subfields within the AC—A1, AAF, and VAF—we analyzed the spatial distribution of the P1 response.

## Neuropixels recordings

For recording neuronal activity across different cortical layers, we used the Neuropixels 1.0 electrode array [63]. The array features 384 active recording sites (out of 960 sites total) located at the bottom approximately 4 mm of an approximately 10 mm shank (70 μm wide, 24 μm thick, 20 μm electrode spacing), with the reference and ground, shorted together. The

probe's ground and reference pads were soldered to a pin, which was then connected to a skull screw on the animal. Recordings were made in internal reference mode.

Data acquisition for the Neuropixels arrays was performed using Neuropixels version 1.0 acquisition hardware, specifically the National Instruments PXIe-1071 chassis and PXI-6133 I/O module for recording analog and digital inputs (Neuropixels version 1.0, IMEC, Belgium). The signal was collected from the most distal 384 recording sites (bank 0) at a 30 kHz sampling rate. Raw voltage traces from the Neuropixels arrays were filtered, amplified, multiplexed, and digitized on-probe. Data acquisition and initial processing were managed using Open Ephys software ([www.open-ephys.org](www.open-ephys.org)). Spike sorting was conducted using Kilosort 3 [64], and further manual curation was performed with Phy software ([https://phy.readthedocs.io/en/latest/](https://phy.readthedocs.io/en/latest/)). This process involved removing artifacts, merging duplicate clusters, and discarding units with insufficient refractory periods, intermingled waveforms, or unstable firing patterns. Only well-isolated neurons meeting strict inclusion criteria were retained for analysis. The cortical depth of the recording site, estimated via CSD analysis, had to be within 0 μm to +1,800 μm from the pial surface. The lower bound of 1,800 μm was chosen to confine recordings to the auditory cortex, preventing the inclusion of neurons from subcortical areas or unintended regions. Additionally, neurons had to fire at least 200 spikes per session to ensure sufficient statistical power, while those exhibiting strong drift in firing rate or waveform shape were excluded to maintain stationarity.

## Characterization of the frequency response area

To assess the frequency-tuning properties of all recorded neurons, we measured the frequency response area (FRA) at each recording site. The auditory stimuli consisted of pure tone bursts with a 5-ms rise/fall time and a 20-ms steady-state duration. The tones covered a wide range of frequencies, spanning 18 distinct values from 1.6 to 64 kHz, and were presented at seven different sound pressure levels, ranging from 20 to 80 dB SPL in 10-dB steps. In total, 126 unique tone bursts were used to characterize the FRA. The stimuli were delivered in a pseudorandom sequence with an inter-tone interval of 600 ms, and each tone was repeated 20 times. FRA maps were generated for each neuron based on their responses to the presented tones. After examining the FRA maps, we selected two tones that were one octave apart and elicited robust responses from most recorded neurons. These two tones were then used as the stimuli for the subsequent experiments investigating the neural responses to tone sequences and omissions using an intensity well above the threshold of a typical neuron.

## Experimental design

To investigate the neural responses to tone sequences and omissions, we designed eight experimental conditions, each employing an oddball paradigm with two distinct tones and a fixed 5% probability of tone omissions. The stimuli were the two sinusoidal tones chosen before with a duration of 50 ms, including 5-ms rise and fall ramps. These tones were arranged in sequences of 1,000 stimuli, with an SOA of 150 ms. In the first condition, one tone (Tone A) comprised 95% of the sequence, while the other tone (Tone B) was absent. Across the subsequent conditions, the proportion of Tone B gradually increased by 5% increments, while the proportion of Tone A decreased accordingly. This resulted in the following sequences:

Sequence 1: 95% Tone A, 0% Tone B, 5% omissions

Sequence 2: 90% Tone A, 5% Tone B, 5% omissions

Sequence 3: 85% Tone A, 10% Tone B, 5% omissions

Sequence 4: 75% Tone A, 20% Tone B, 5% omissions

Sequence 5: 50% Tone A, 45% Tone B, 5% omissions

Sequence 6: 20% Tone A, 75% Tone B, 5% omissions

Sequence 7: 10% Tone A, 85% Tone B, 5% omissions

Sequence 8: 0% Tone A, 95% Tone B, 5% omissions

This design allowed us to systematically manipulate the relative probabilities of the two tones while maintaining a consistent rate of omissions. By progressively shifting the balance between the tones across conditions, we could examine how the neural responses to tone omissions were influenced by the statistical context provided by the sequence.

### Data analysis

**Omission responses.** Omission response was defined as the firing rate measured in a 5–120 ms post-omission window, with a local baseline subtracted on a per-trial basis. The baseline was computed as the mean firing rate within a −24 ms to +5 ms window relative to the expected tone onset, a period that closely reflects the neuron's state immediately prior to the omitted tone and controls for transient adaptations or fluctuations that might occur mid-sequence. The computed baseline was then subtracted from the mean firing rate obtained during the post-omission period, yielding a final omission response value that was typically around zero when the neuron exhibited no net change in firing rate. Although we corrected responses using this immediate pre-omission baseline, we also examined pre-sequence spontaneous firing rates and observed no correlation between those rates and omission responses (see S8 Fig). This methodology is consistent with established work [32], which demonstrated robust omission responses using a similar immediate reference period.

**Clustering of correlation coefficients.** To examine the presence of distinct groups based on the strength of the correlation between each neuron's omission response and predictability of $O_P$ tone, we performed $k$-means clustering on the correlation coefficients. Thus, neurons encoding predictability exhibited either strong positive or strong negative correlations. To determine the optimal number of clusters, we used the elbow method, which identifies the "knee point" in the plot of within-cluster sum of squares (WCSS) against the number of clusters. The knee point suggests the optimal number of clusters where adding more clusters does not significantly improve the model's fit to the data (see S1 Fig). The $k$-means clustering analysis was performed using the $k$-means function in MATLAB with 10 replicates for each cluster number. The WCSS was calculated for each cluster number, and the optimal number of clusters was determined by finding the knee point in the WCSS plot. The clustering results were then visualized using histograms to display the distribution of correlation coefficients within each cluster.

**Split-Half approach for PEON classification.** To mitigate potential biases such as double dipping—where the same data are used both to select and evaluate effects—and to ensure the robustness of our probability encoding analysis, we employed a split-half approach. For each probability condition, the 50 omission trials were divided into two equal halves—an odd-indexed "training set" and an even-indexed "testing set," with each set providing 25 omission trials per condition. Using only the training set, we computed Spearman's rank correlation between each neuron's omission response and the probability condition. Neurons with a significant correlation ($p < 0.05$) were provisionally labeled as "correlated," and the $O_P$ tone for each neuron was defined based on the direction of the correlation in the training data. From this subset of correlated neurons, we then pooled the training-set trials in the conditions where the $O_P$ was standard (75%–95%) and applied a one-sample Wilcoxon sign-rank test ($p < 0.05$) to determine whether their omission response was significantly above zero. Because the omission response is computed as the firing rate minus a local baseline, testing against zero effectively assesses whether post-omission activity exceeds the neuron's immediate pre-omission baseline. Only neurons that exhibited both a significant correlation and a robust omission response were designated as PEONs in the training set. All subsequent population plots—such as boxplots or group correlations—were generated solely from the testing set, ensuring that any observed trends were unbiased by the training-set selection process. Complementary analyses were also carried out using even-indexed trials for training (with odd-indexed trials for testing) as well as a non-split method utilizing all trials; these additional analyses, presented in the in S2–S5 Figs, confirm that our main conclusions do not depend on the specific trial split.

**Data subsampling of tone responses.** To ensure that neuronal responses across different tone probability conditions were not biased by differences in statistical power due to varying trial counts, we implemented a standardized subsampling procedure. For each neuron, within each condition, we limited the analysis of tones to 50 trials per stimulus type to prevent conditions with more trials from exerting disproportionate influence. If more than 50 trials were available, we selected every $k$-th trial, where $k$ was chosen to achieve 50 trials (e.g., for 100 trials, $k = 2$; for 200 trials, $k = 4$, etc.). This ensured that all conditions were analyzed with equal statistical power.

For analyses comparing tone responses with the split-half approach used for PEON identification and characterization (see "PEON Identification" section), an additional subsampling step was performed. After the initial reduction to 50 trials per condition, the remaining trials were further divided into ODD and EVEN subsets, with each subset matched to the corresponding half used for omission responses.

**Buildup of omission responses.** For each PEON and each repetition, omission responses were extracted from the four conditions in which the $O_P$ tone was presented as the standard (i.e., 75%, 85%, 90%, and 95% $O_P$ tone). The omission responses were binned based on their sequence position, and for each bin the mean firing rate and standard error were computed, yielding a two-dimensional response matrix that captures the temporal evolution of omission responses.

The mean omission response was then modeled with a logistic growth function using MATLAB's *nlinfit* function. The model was defined as: $FR(n) = a/[1 + \exp(-b \times (n - c))]$, where $n$ represents the bin center corresponding to the sequence position, and $a$, $b$, and $c$ are free parameters estimated from the data. The goodness-of-fit was assessed by calculating the $R$-squared value based on the residual and total sums of squares. To control for non-specific drift in neural activity, responses to standard tones during the first 50 trials were similarly analyzed. These responses were fitted with an exponential decay function of the form: $FR(n) = d + e \times \exp(-f \times n)$, where $n$ denotes the trial number, and $d$, $e$, and $f$ are free parameters. The fit quality was evaluated using the $R$-squared metric.

**Laminar and areal distribution of neurons.** To determine the laminar distribution of neurons, we adapted the methodology from Shiramatsu and colleagues [62]. We used CSD analysis on local field potentials recorded with Neuropixels electrodes to identify cortical layer boundaries. The earliest clear current sink, typically corresponding to the border between layers 3 and 4, was used as a key marker. Individual neuron depths were estimated based on their distance from the probe tip, as determined by spike sorting. Neurons were then classified into supragranular (Layers 1–3), granular (Layer 4), and infragranular (Layers 5–6) layers. Cortical thickness and layer boundaries were set according to Shiramatsu and colleagues [62].

**Tone and omission selectivity indices.** To assess the relationship between tone-evoked selectivity and omission selectivity, we computed selectivity indexes – TSI and OSI for each neuron. TSI was calculated as:

$$TSI = \frac{T_{O_P} - T_{O_{NP}}}{T_{O_P} + T_{O_{NP}}}$$

$T_{O_P}$ and $T_{O_{NP}}$ represent the neuron's mean firing rate in response to the $O_P$ and $O_{NP}$ tones, respectively, averaged across the seven probability conditions in which each tone was presented. OSI was calculated as:

$$OSI = \frac{O_{O_P} - O_{O_{NP}}}{O_{O_P} + O_{O_{NP}}}$$

$O_{O_P}$ and $O_{O_{NP}}$ denote the neuron's mean firing rate in response to omissions occurring in the 4 sequences where the $O_P$ and $O_{NP}$ tones were standard, respectively. To test whether a neuron's tone selectivity predicted its omission selectivity, we computed Spearman's correlation between TSI and OSI.

**Computational modeling.** We constructed a circuit of leaky integrate-and-fire neurons following

$$\tau_m \frac{dV}{dt} = -V + R_m \left( I_{exc} - I_{inh} + I_{pred} + I_{stim} \right)$$

with membrane time constant $\tau_m = 30$ ms and membrane resistance $R_m = 1$ MΩ. When $V > 10$ mV, neurons emitted a spike ($X = 1$) and were reset to $V = 0$ mV for a refractory period of 2 ms. Synaptic currents $I_{exc}$ and $I_{inh}$ were modeled with instantaneous rise and exponential decay,

$$\tau_{exc} \frac{dI_{exc}(t)}{dt} = -I_{exc}(t) + wX(t - d)$$

and equivalently for $I_{inh}$, with time constants $\tau_{exc} = 5$ ms and $\tau_{inh} = 20$ ms. Spikes from presynaptic neurons $X \in \{0, 1\}$ were delivered with a synapse-specific delay $d$ and weight $w$ as detailed in S1 Table within each stream module. Lateral connections between streams $s_1$ and $s_2$ delivered spikes with delay $d = 5 \left| s_2 - s_1 \right|$ ms and weight $w = \frac{1.2\ \mu A}{(s_2 - s_1)^2}$. Sensory stimulation was presented through the input neuron I by setting its $I_{stim}$ to a constant $S\_strength = 0.2$ μA (target stream) or $S\_strength = 0.1$ μA (adjacent streams) for 20 ms. Similarly, prediction signals were fed to the P neuron by setting $I_{pred} = P\_strength * 0.2$ μA with identical timing. $P\_strength$ was initialized to 0 and updated by spikes in prediction neurons as follows:

$$\frac{d}{dt} P\_strength = 0.001 X_{PE+} - 0.0003 X_{PE-}$$

and clipped to ensure $0 \leq P\_strength \leq 1$ at all times. Both $I_{stim}$ and $I_{pred}$ were raised to their respective values for 20 ms at the start of each trial, then reset to $0$ μA for the remaining 130 ms.

   The model was written in Brian2 [65], which allowed calculating exact solutions for the internal neurons of the circuit (PE+, PE−, I+, I−). The remaining neurons (P and I), synapses and $P\_strength$ were integrated with a time step of 0.1 ms by forward Euler integration. Simulations were run over 500 trials.

   **Sensory adaptation.** To model sensory adaptation, we adjusted the input $I_{stim}$ by using an adaptation factor $a$, which is greater than zero, and a base firing factor $b$, which is between 0 and 1. The input strength for the $A$ and $B$ streams under different values of Prob(A) was adjusted as follows:

$$S_{strength_A} = (1 - b) * e^{\frac{Prob(A)}{a}} + b$$

$$S_{strength_B} = (1 - b) * e^{(1 - Prob(A))/a} + b$$

Examples of $S\_strength_A$ versus Prob(A) are shown in S6 Fig, with 12 combinations of $a$ (0.01, 0.05, 0.1, 0.5) and $b$ (0.1, 0.3, 0.5).

## Supporting information

**S1 Fig. Elbow analysis for optimal cluster selection. (A)** The elbow method was used to determine the optimal number of clusters for classifying neurons based on the correlation between omission response and $O_P$ tone probability. The x-axis represents the number of clusters, while the y-axis represents the WCSS, which measures clustering compactness. A sharp inflection point, or "elbow," is observed at $k=2$, indicating that a two-cluster solution best balances model complexity and explained variance. This supports the classification of neurons into two distinct response groups. (PDF)

**S2 Fig. PEON classification and probability encoding across ODD, EVEN, and ALL Trials. (A)** left: Classification of PEON$_{EVEN}$. Pie chart, illustrating how neurons identified in odd trials were categorized when evaluated on even trials. Neurons are grouped based on whether they exhibit no correlation with tone probability, correlation but no omission response, or both correlation and a measurable omission response ("PEON$_{EVEN}$"). Underlying data are provided in S3 Data (sheet "S2A_Individual_Points"). **(A)** right: Box plot summarizing omission responses for PEON$_{EVEN}$, tested on ODD trials. The x-axis represents the O$_P$ tone probability (0%–95%), and the y-axis indicates mean baseline-subtracted firing rate in response to the omission, calculated over a 5–120 ms window after the expected tone onset. Boxes correspond to the IQR with a horizontal line marking the median. Whiskers extend to 1.5× IQR. Individual data points are shown as over-laid dots, with means at each probability level marked by red diamonds connected by a dashed line. The text annotation shows the Spearman's rank correlation coefficient (rho) and associated p-value, quantifying the strength and statistical significance of the monotonic relationship between probability and neural response. **(B)** left: Pie chart showing neuron categorization when using all available trials without splitting into odd or even subsets. Underlying data are provided in S3 Data (sheet "S2B_Individual_Points") **(B)** right: Box plot displaying omission responses for PEONs identified using all trials, allowing direct comparison with split-data analyses.
(PDF)

**S3 Fig. Laminar distribution of PEON$_{EVEN}$ and PEON$_{ALL}$ (A)** Laminar distribution of PEON$_{EVEN}$. The distribution of the recorded population (gray bars) and the neurons classified as PEON (blue bars) is shown as a function of cortical depth (y-axis). Horizontal dashed lines indicate the approximate boundaries between supragranular (0–600 µm), granular (600–900 µm), and infragranular (900–1,400 µm) layers, based on current source density analysis. **(B)** Laminar distribution of PEON$_{ALL}$. Same as **(A)**, but for PEON$_{ALL}$, showing the depth distribution when using all available trials without splitting into odd and even subsets.
(PDF)

**S4 Fig. Trial-by-trial dynamics of omission responses and standard tone adaptation in PEON$_{EVEN}$ and PEON$_{ALL}$.**
**(A)** Left: Firing rate of PEON$_{EVEN}$ for omission events (on ODD trials) in the four conditions where the O$_P$ tone was the standard (75%, 85%, 90%, 95%), grouped by sequence position. The x-axis represents the bin center (i.e., sequence position), and the y-axis represents the firing rate in spikes per second. The black dots indicate the mean firing rate across all PEON$_{EVEN}$ for each bin, and the error bars represent SEM, showing variability across neurons. The blue line represents the logistic growth curve fit to the data, illustrating how omission responses evolve over successive standards. Right: Firing rate of PEON$_{EVEN}$ in response to the presentation of tones across the first 50 trials. The x-axis represents the trial number, and the y-axis represents the firing rate in spikes per second. The black dots indicate the mean firing rate across all PEON$_{EVEN}$ for each trial, and the error bars represent SEM, showing variability across neurons. The blue line represents the exponential decay model fit to the data, indicating a rapid decline in response intensity over the initial trials as the neurons adapt to the repeated tone presentations. **(B)** Left: Firing rate of PEON$_{ALL}$ for omission events. Same as (**A**, left), but for PEON$_{ALL}$, showing the omission response buildup when using all available trials without splitting into odd and even subsets. Right: Firing rate of PEON$_{ALL}$ in response to tone presentations. Same as (**A**, right), but for PEON$_{ALL}$, showing the adaptation to standard tones across the first 50 trials using all available trials.
(PDF)

**S5 Fig. Omission selectivity in PEON$_{EVEN}$ and PEON$_{ALL}$ (A)** Left: Pie chart categorizing 99 PEON$_{EVEN}$ neurons (identified on even trials and validated on odd trials) into omission non-selective (46) and omission selective (53). Right: Among the 53 omission-selective neurons, 5 responded exclusively to the O$_P$ tone (O$_P$-only), 12 responded exclusively to the O$_{NP}$ tone (O$_{NP}$-only), 6 did not significantly respond to either tone (None), and 30 responded to both tones (Both). **(B)** Left: Pie chart categorizing 200 PEON$_{ALL}$ neurons into omission non-selective (59) and omission selective (141). Right: Among the

141 omission-selective neurons, 11 responded exclusively to $O_P$ ($O_P$-only), 24 to $O_{NP}$ ($O_{NP}$-only), 23 to neither tone (None), and 83 to both (Both).
(PDF)

**S6 Fig. Model comparisons with adapted sensory signals. (A)** Firing rates of thalamic neurons ($n = 130$) for Tone 1 (red) and Tone 2 (blue) across varying probabilities of the preferred tone (Prob($O_P$)). Error bars indicate SEM. **(B)** To mimic the thalamic sensory inputs shown in the panel A, the strength of sensory signals (*S_strength*) for Tones A (red) and B (blue) was modelled as exponential decay functions. Simulations with various combinations of the adaptation factor *a* and base firing factor *b* were performed, and the results are shown with different symbols. **(C)** Comparison of mean firing rates in different model configurations: Lateral PE+ to PE– (left), Lateral I to PE– (middle), and No lateral (right). Firing rates for Tones A (red), B (blue), and omissions (O, black) are plotted against Prob(A). The same symbols are used as in panel B.
(PDF)

**S7 Fig. Positive error encoding in non-PEONs. (A)** Population-level firing rates comparing omission responses (black), tone responses to the $O_P$ tone (blue), and tone responses to the $O_{NP}$ tone (red), plotted against the probability of the $O_P$ tone. The *y*-axis indicates firing rate (spikes/s), and error bars show standard error of the mean. All data are drawn from the EVEN trials of non-PEONs identified in ODD trials.
(PDF)

**S8 Fig. Pre-sequence spontaneous firing rate versus omission response strength. (A)** Scatterplot of pre-sequence spontaneous firing rate versus omission response strength. Each point represents a single neuron in a specific probability condition. The *x*-axis shows the pre-sequence spontaneous firing rate (spikes/s), measured before the onset of tone sequences, while the *y*-axis indicates the omission response strength (spikes/s). The Pearson correlation coefficient (*r*) and *p*-value are displayed in the lower-right corner. Underlying data are provided in S4 Data.
(PDF)

**S1 Data. Fig 2D source data.** Sheet "Individual_Points": baseline-subtracted omission responses for each PEONODD neuron on EVEN trials (one row per neuron × probability). Sheets "Overall_Means", "Statistics": group means ± SEM and Spearman ρ, *p*.
(XLSX)

**S2 Data. Fig 4A source data.** Sheet "Individual_Points": raw firing-rate values for every PEONODD neuron (columns: NeuronID, Probability_Percent, ResponseType [Omission | TonePref | ToneNonPref], FiringRate). Sheet "Response_Data": group means ± SEM for the three curves. Sheet "Correlations": Spearman ρ, *p* for each response type.
(XLSX)

**S3 Data. S2A and S2B Fig source data.** Sheet "S2A_Individual_Points": raw omission-response values used in S2A Fig. Sheet "S2B_Individual_Points": probability–response pairs used in S2B Fig.
(XLSX)

**S4 Data. S8 Fig source data.** Sheet "Data_Points": pre-sequence spontaneous firing rate and omission-response strength for each neuron. Sheet "Statistics": Pearson *r* and *p* for the correlation shown in S8 Fig.
(XLSX)

**S1 Text. Mathematical derivation of lateral prediction suppression from the free energy principle.** This supplement provides the mathematical framework discussed in the main text under "Lateral prediction suppression and the free energy principle." It formally derives how suppressive lateral interactions between sensory streams are a natural consequence of applying the free energy principle to a simplified auditory model.
(DOCX)

**S1 Table. Synaptic parameters for the intra-stream predictive-coding module.** This table contains the core parameters for the intra-stream connections of the neural circuit model (5A Fig).
(DOCX)

## Acknowledgments

We thank the International Research Center for Neurointelligence at the University of Tokyo for support and promotion of collaboration. We are also grateful to the members of the Chao and Takahashi laboratories for valuable feedback, technical assistance, and helpful discussions throughout the course of this work.

## Author contributions

**Conceptualization:** Amit Yaron, Zenas C. Chao.

**Data curation:** Amit Yaron, Zenas C. Chao.

**Formal analysis:** Amit Yaron, Felix B. Kern, Kenichi Ohki, Zenas C. Chao.

**Funding acquisition:** Amit Yaron, Tomoyo Shiramatsu-Isoguchi, Hirokazu Takahashi, Zenas C. Chao.

**Investigation:** Amit Yaron, Tomoyo Shiramatsu-Isoguchi.

**Methodology:** Amit Yaron, Tomoyo Shiramatsu-Isoguchi, Zenas C. Chao.

**Resources:** Tomoyo Shiramatsu-Isoguchi, Hirokazu Takahashi.

**Software:** Amit Yaron, Felix B. Kern, Zenas C. Chao.

**Supervision:** Hirokazu Takahashi, Zenas C. Chao.

**Validation:** Amit Yaron, Zenas C. Chao.

**Visualization:** Amit Yaron, Felix B. Kern, Zenas C. Chao.

**Writing – original draft:** Amit Yaron, Kenichi Ohki, Zenas C. Chao.

**Writing – review & editing:** Amit Yaron, Tomoyo Shiramatsu-Isoguchi, Felix B. Kern, Hirokazu Takahashi, Zenas C. Chao.

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
