## [Editor Report · Decision Letter 0]

Dear Dr Chao, 

Thank you for submitting your manuscript entitled "Probability Encoding Omission Neurons in Auditory Cortex Signal Negative Prediction Error" for consideration as a Research Article by PLOS Biology.

Your manuscript has now been evaluated by the PLOS Biology editorial staff and I am writing to let you know that we would like to send your submission back for external peer review.

Once your full submission is complete, your paper will undergo a series of checks in preparation for peer review. After your manuscript has passed the checks it will be sent out for review. To provide the metadata for your submission, please Login to Editorial Manager (https://www.editorialmanager.com/pbiology) within two working days, i.e. by Mar 16 2025 11:59PM.

Kind regards,

Christian

Christian Schnell, PhD

Senior Editor

PLOS Biology

cschnell@plos.org

---

## [Decision Letter · Decision Letter 1]

Dear Dr Chao,

Thank you for your patience while we considered your revised manuscript "Probability Encoding Omission Neurons in Auditory Cortex Signal Negative Prediction Error" for consideration as a Research Article at PLOS Biology. Your revised study has now been evaluated by the PLOS Biology editors, the Academic Editor and two of the original reviewers.

In light of the reviews, which you will find at the end of this email, we are pleased to offer you the opportunity to address the remaining points from the reviewers in a revision that we anticipate should not take you very long. We will then assess your revised manuscript and your response to the reviewers' comments with our Academic Editor aiming to avoid further rounds of peer-review, although we might need to consult with the reviewers, depending on the nature of the revisions.

**IMPORTANT - SUBMITTING YOUR REVISION**

*Resubmission Checklist*

*Published Peer Review*

*PLOS Data Policy*

*Blot and Gel Data Policy*

Sincerely,

Christian

Christian Schnell, PhD

Senior Editor

PLOS Biology

cschnell@plos.org

REVIEWS:

Reviewer #2: It is commendable that the authors made a serious effort to address our previous concerns. However, regarding the main concern of double dipping, we are not sure this is fully addressed. In their response the authors state that "Trials were divided into two independent subsets—odd and even trials—where one subset was used to classify PEONs and the other was used to quantify omission responses. This prevents circularity by ensuring that selection and characterization are independent." - this would be correct. In the paper, however, it is less clear whether this is implemented systematically. E.g. (line 178): "These findings demonstrate the consistency of the PEON population regardless of which trial subset was used." If this is based on figure 2C, this conclusion is not warranted as the authors would need to show that the PEONodd and PEONeven are the same population of neurons (or at least that the overlap is above chance). In Figure 2D, it appears implemented correctly (but something is amiss with the statistics - should p=1.6x10^27 read p=1.6x10^-27 - and even that does not appear to match the data shown?). The authors should recheck their analysis to confirm that they do not use comparisons of the variety shown in Figure 2C (and fix, or remove Figure 2C). 

Reviewer #3: The authors have addressed all my comments.

---

## [Editor Report · Decision Letter 2]

Dear Dr Chao,

Thank you for your patience while we considered your revised manuscript "Probability Encoding Omission Neurons in Auditory Cortex Signal Negative Prediction Error" for publication as a Research Article at PLOS Biology. This revised version of your manuscript has been evaluated by the PLOS Biology editors and the Academic Editor.

Based on our Academic Editor's assessment of your revision, we are likely to accept this manuscript for publication, provided you satisfactorily address the following data and other policy-related requests:

* We would like to suggest a different title to improve its accessibility for our broad audience: 

Auditory cortex neurons that encode negative prediction errors respond to omissions of sounds in a predictable sequence

* Please add the links to the funding agencies in the Financial Disclosure statement in the manuscript details.

* DATA POLICY:

Regardless of the method selected, please ensure that you provide the individual numerical values that underlie the summary data displayed in the following figure panels as they are essential for readers to assess your analysis and to reproduce it: 2D, 4A, S2AB and S8

* CODE POLICY

We expect to receive your revised manuscript within two weeks. 

*Published Peer Review History*

*Press*

Sincerely,

Christian

Christian Schnell, PhD

Senior Editor

cschnell@plos.org

PLOS Biology

---

## [Editor Report · Decision Letter 3]

Dear Dr Chao,

Thank you for the submission of your revised Research Article "Auditory cortex neurons that encode negative prediction errors respond to omissions of sounds in a predictable sequence" for publication in PLOS Biology. On behalf of my colleagues and the Academic Editor, Jennifer Bizley, I am pleased to say that we can in principle accept your manuscript for publication, provided you address any remaining formatting and reporting issues. These will be detailed in an email you should receive within 2-3 business days from our colleagues in the journal operations team; no action is required from you until then. Please note that we will not be able to formally accept your manuscript and schedule it for publication until you have completed any requested changes.

PRESS

Sincerely, 

Christian

Christian Schnell, PhD

Senior Editor

PLOS Biology

cschnell@plos.org